Li *et al. Genome Biology*      (2023) 24:68

**RESEARCH**

# Large-scale analysis of de novo mutations identifies risk genes for female infertility characterized by oocyte and early embryo defects

Qun Li[1,2†], Lin Zhao[1†], Yang Zeng[1†], Yanping Kuang[3†], Yichun Guan[4†], Biaobang Chen[5†], Shiru Xu[6], Bin Tang[7], Ling Wu[3], Xiaoyan Mao[3], Xiaoxi Sun[8], Juanzi Shi[9], Peng Xu[10], Feiyang Diao[11], Songguo Xue[12], Shihua Bao[13], Qingxia Meng[14], Ping Yuan[15], Wenjun Wang[15], Ning Ma[16], Di Song[17], Bei Xu[18], Jie Dong[1], Jian Mu[1], Zhihua Zhang[1], Huizhen Fan[1], Hao Gu[1], Qiaoli Li[1], Lin He[19], Li Jin[20], Lei Wang[1*] and Qing Sang[1*]

†Qun Li, Lin Zhao, Yang Zeng, Yanping Kuang, Yichun Guan, and Biaobang Chen contributed equally.

*Correspondence:
wangleiwanglei@fudan.edu.cn;
sangqing@fudan.edu.cn

[1] Institute of Pediatrics, Children's Hospital of Fudan University, the Shanghai Key Laboratory of Medical Epigenetics, the Institutes of Biomedical Sciences, the State Key Laboratory of Genetic Engineering, Fudan University, Shanghai 200032, China
Full list of author information is available at the end of the article

## Abstract

**Background:** Oocyte maturation arrest and early embryonic arrest are important reproductive phenotypes resulting in female infertility and cause the recurrent failure of assisted reproductive technology (ART). However, the genetic etiologies of these female infertility-related phenotypes are poorly understood. Previous studies have mainly focused on inherited mutations based on large pedigrees or consanguineous patients. However, the role of de novo mutations (DNMs) in these phenotypes remains to be elucidated.

**Results:** To decipher the role of DNMs in ART failure and female infertility with oocyte and embryo defects, we explore the landscape of DNMs in 473 infertile parent–child trios and identify a set of 481 confident DNMs distributed in 474 genes. Gene ontology analysis reveals that the identified genes with DNMs are enriched in signaling pathways associated with female reproductive processes such as meiosis, embryonic development, and reproductive structure development. We perform functional assays on the effects of DNMs in a representative gene Tubulin Alpha 4a (*TUBA4A*), which shows the most significant enrichment of DNMs in the infertile parent–child trios. DNMs in *TUBA4A* disrupt the normal assembly of the microtubule network in HeLa cells, and microinjection of DNM *TUBA4A* cRNAs causes abnormalities in mouse oocyte maturation or embryo development, suggesting the pathogenic role of these DNMs in *TUBA4A*.

**Conclusions:** Our findings suggest novel genetic insights that DNMs contribute to female infertility with oocyte and embryo defects. This study also provides potential genetic markers and facilitates the genetic diagnosis of recurrent ART failure and female infertility.

**Keywords:** Female infertility, De novo mutations, Reproductive pathways, *TUBA4A*, Microtubule stability

## Background

Female infertility has become a global health issue affecting over 10% of all reproductive-age women worldwide. Although assisted reproductive technology (ART) helps many infertile women conceive and give birth [1], many infertile women suffer recurrent ART failure due to different phenotypes including oocyte maturation arrest and early embryo arrest. It has been demonstrated that these phenotypes show Mendelian inheritance patterns, and several monogenic mutations have been reported [2–4]. However, previous studies have mainly focused on inherited mutations and can only explain a limited number of affected individuals, and for the majority of corresponding patients, the underlying genetic factors remain to be elucidated.

De novo mutations (DNMs), which occur in individuals and not in their parents, are an important source of genetic variation. Unlike inherited mutations, DNMs are exposed to less selective pressure and are associated with human evolution and diseases [5–8]. With the development of whole-exome sequencing technologies, considerable efforts have been made to establish a causal relationship between DNMs and human diseases such as neurodevelopmental disorders [9, 10], heart disease [11], and early-onset high myopia [12]. The fundamental effects of DNMs in male infertility have recently been reported [13], but the landscape and role of DNMs in female infertility with oocyte and embryo defects remain to be elucidated.

To address the role of DNMs in female infertility with ART failure, we analyzed whole exome sequencing data from 473 infertile parent–child trios and identified 481 DNMs. These mutant genes were significantly associated with female reproductive development pathways based on gene expression dynamics. We further performed functional assays on the effects of DNMs in Tubulin Alpha 4a (*TUBA4A*), which showed the most significant enrichment of rare DNMs in our infertile parent–child trios. Protein structure prediction and immunofluorescence in HeLa cells revealed that *TUBA4A* mutations resulted in microtubule instability. Microinjecting *TUBA4A* mutant cRNAs into mouse oocytes led to reduced rates of oocyte maturation or to disruptions in embryo development, thus mimicking the phenotypes of infertile individuals. Taken together, DNMs contributed significantly to the occurrence of female infertility characterized by abnormality in oocyte maturation and embryonic development, and *TUBB4A* appears to be a novel pathogenic gene in infertile women with oocyte or embryo problems. This study will provide novel insights into the genetic etiology of female infertility with oocyte and embryo defects.

## Results

### Landscape of DNMs in infertile women with ART failure

To clarify the role of DNMs in ART failure and female infertility with oocyte and embryo defects, we performed DNM analysis using whole-exome sequencing data of 473 infertile parent–child trios with recurrent failure of IVF/ICSI attempts due to oocyte maturation arrest and embryo arrest (Fig. 1; Additional file 2: Table S1). None of these trios had been diagnosed with a recognized genetic disease. Whole-exome sequencing data from 92 trios of unaffected siblings were analyzed as an in-house control. There was no appreciable genetic population stratification in our samples according to the PCA analysis (Additional file 1: Fig. S1a), and the average sequencing depth was 138 (Additional file 1:

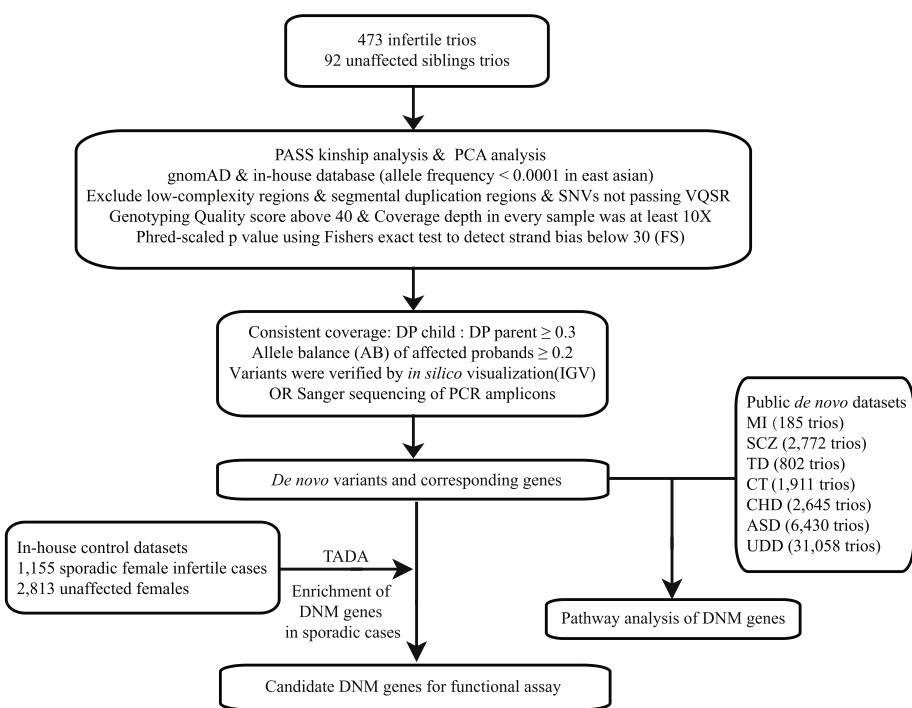

**Fig. 1** Pipeline of data processing in this study. Sample- and variant-based quality control that was applied to the raw GVCF to generate the DNMs used in downstream analyses. Abbreviations: GVCF, genomic variant call format; PCA, principal component analysis; VQSR, variant quality score recalibration; FS, Fisher strand; DP, read depth; TADA, transmission and de novo association analysis; IGV, integrative genomics viewer

Fig. S1b). Altogether, 481 high-confidence DNMs were generated exome-wide from 473 infertile patient trios with an average of 1.02 DNMs per case (Additional file 2: Table S2), while 69 DNMs were identified in 92 unaffected sibling trios. The number of DNMs in patient and sibling trios all closely fit the Poisson distribution (Fig. 2a, Additional file 1: Fig. S1c), and the average age in our patient cohort is 31 (Additional file 1: Fig. S1d).

According to the potential effects of mutations on protein function, we divided the list of DNMs into three classes: synonymous, missense, and loss-of-function (LOF). Except for 0.62% of non-frameshifting mutations, most of the DNMs (61.6%) were missense mutations, 12.53% were LOF, and the remaining 25.25% were synonymous (Fig. 2b). Of all the missense mutations, 29.57% were predicted to be damaging and 11.09% were possibly damaging (Fig. 2b). Next, we calculated the enrichment of different mutations in patient, unaffected sibling trios, and public trios comprising 1011 unaffected females [10]. There was a significant excess of LOF DNMs in the patient group (enrichment$= 1.47$; $p = 2.5 \times 10^{-3}$) but not in siblings (enrichment$= 0.746$; $p = 0.813$) and public controls (enrichment$= 1.07$; $p = 0.255$) (Fig. 2c).

Next, to estimate the number of DNM genes contributing to the disease phenotypes based on our collected trios, we utilized a previously established maximum likelihood estimation (MLE) method based on recurrent DNMs and sample size [14–16]. Finally, our data fit best with a model of 419 risk-associated genes (Fig. 2d), suggesting that approximately 419 DNM genes may contribute to the corresponding phenotypes.

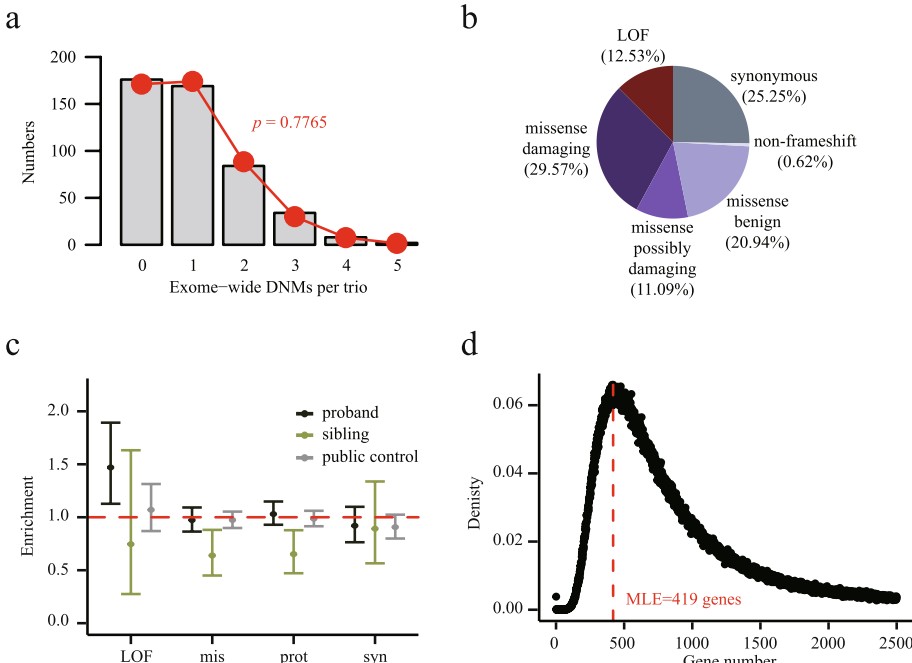

**Fig. 2** De novo mutations are enriched in individuals with female infertility. **a** Distribution of coding DNMs per trio in probands. **b** Distribution of DNMs by mutation class for all infertile parent-child trios. **c** Enrichment of DNMs (95% confidence interval) by mutation class for probands and siblings. Mutation classes were labeled as the following terms: LOF (Loss-of-Function), mis (missense), prot (protein-altering = mis + LOF), and syn (synonymous). Expected number of DNMs was calculated by denovolyzeR and enrichment was calculated by observed number/expected number. *P* value and confidence interval were calculated by comparing the observed value to the expected value with Poisson test. **d** Maximum likelihood estimate (MLE) of risk genes. We estimated that 419 genes contribute to female infertility risk based on vulnerability to damaging DNMs

### DNM genes are involved in pathways associated with female reproduction

To explore the functional role of DNMs, we performed GO analysis ($p < 0.01$) using stage-specific genes according to integrated expression clusters of different human early reproductive stages (Additional file 2: Table S3) and GO analysis is an established method in analyzing association relationship between DMNs with human diseases [10, 17, 18]. Expression clusters in female FGCs (fetal germ cells) were performed by re-analyzing RNA-seq datasets of different stages, including early FGC, pre-meiotic FGC, meiotic FGC, and dictyate oocyte [19] (Additional file 1: Fig. S2 and Fig. S3). For human folliculogenesis, human matured oocytes, and early embryo development, the developmental stages were referred to according to published results [20–22].

Selected Gene Ontology (GO) terms were integrated into 16 female reproductive stages (Fig. 3). We assigned GO terms to three main categories: meiosis, embryonic development, and reproductive structure development. 12 DNM genes were enriched in pathways related to meiosis, such as meiotic cell cycle ($p = 1.85 \times 10^{-4}$), spindle organization ($p = 1.66 \times 10^{-3}$), and meiotic chromosome segregation ($p = 1.29 \times 10^{-4}$) (Additional file 1: Fig. S3). A total of 29 DNM genes were enriched in pathways related to embryonic development, such as in utero embryonic development ($p = 2.81 \times 10^{-4}$), regulation of translational initiation ($p = 3.81 \times 10^{-3}$) [21],

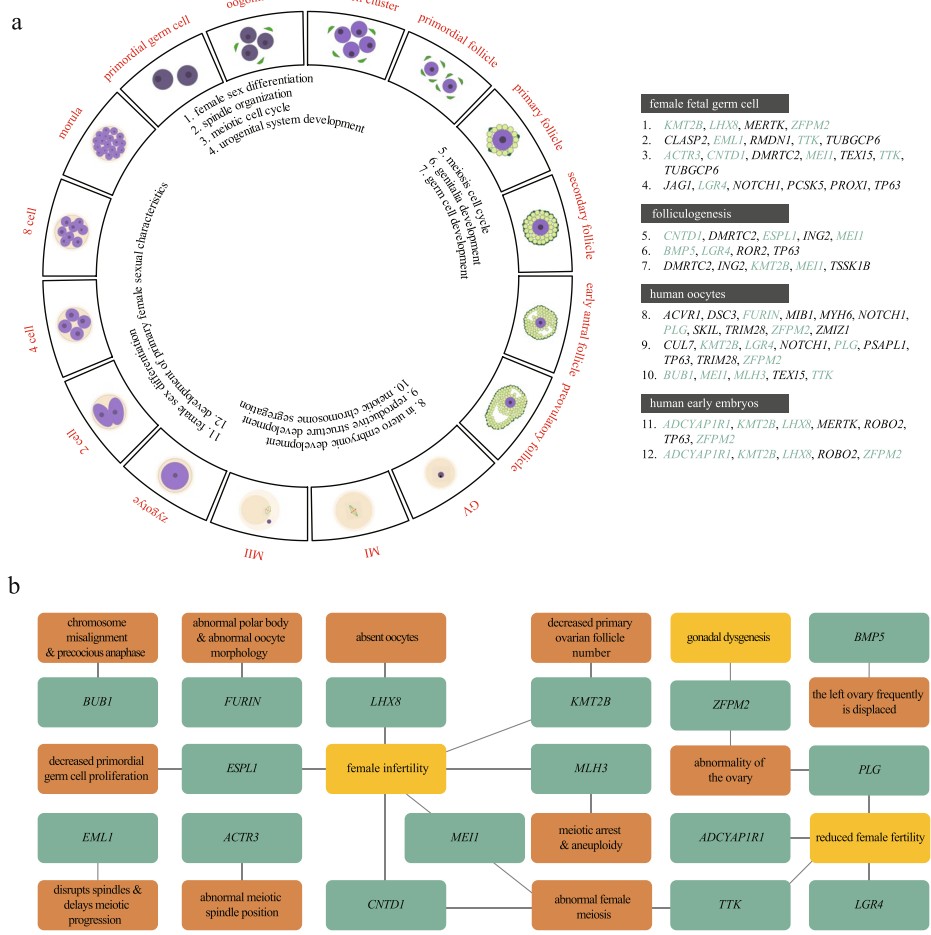

**Fig. 3** DNM genes were involved in pathways associated with female reproduction. **a** Clustering of dynamic gene expression was performed on female germ cells, including female fetal cells, oocytes during folliculogenesis, mature oocytes, and early embryos. Female germ cells were subdivided into sixteen stages, and GO analysis was performed using stage-specific genes. Significantly enriched GO terms (*p* < 0.01) were visualized at sixteen stages of female germ cells, and the pathways and their associated genes are listed on the right (genes associated with female reproductive defects were labeled as green). **b** Network connections of representative mouse genome informatics (MGI) and PubMed phenotypic annotations (green: genes associated with female reproductive defects; orange: female infertile phenotype; brown: description of germ cell defect)

regulation of histone modification ($p = 5.37 \times 10^{-4}$) [23–25], and positive regulation of Wnt signaling pathway ($p = 4.14 \times 10^{-6}$) [26–28] (Additional file 1: Fig. S3). 18 DNM genes were enriched in pathways related to reproductive structure development, such as urogenital system development ($p = 1.11 \times 10^{-3}$), genitalia development ($p = 1.50 \times 10^{-4}$), reproductive structure development ($p = 9.84 \times 10^{-3}$), female sex differentiation ($p = 3.24 \times 10^{-4}$), and development of primary female sexual characteristics ($p = 5.79 \times 10^{-3}$) (Additional file 1: Fig. S3). Overall, a group of DNM genes from our infertile parent–child trios was enriched in pathways of female reproductive processes (Additional file 2: Table S4) and these genes were closely related to female infertility with the characteristics of oocyte and early embryo defects by looking up MGI database and PubMed (Fig. 3b; Additional file 2: Table S5).

To confirm the specific enrichment of female reproductive pathways for DNM genes in our infertile parent–child trios, we performed parallel GO analysis of DNM genes in three control groups, including 92 of our in-house unaffected siblings and 1011 unaffected female individuals [10, 29] and 1097 females with autism spectrum disorder [10] from public databases. No significant enrichment of DNM genes in pathways related to female reproduction was observed in any of the three control groups (Additional file 1: Fig. S4; Additional file 2: Table S3), thus demonstrating the specific contributions of DNM genes to female infertility with oocyte and embryo defects in our infertile parent–child trios.

### TUBA4A showed the most enrichment of mutations in our cohort

We next focused on genes with multiple (two or more) damaging DNMs. Based on previously published thresholds [14, 30–32], four genes were selected for further evaluation, which included *TUBA4A*, *UBQLN1*, *HTR2C*, and *ZFPM2* (Fig. 4a). In addition, these four DNM genes were only enriched in our cohort with oocyte and early embryo defects compared to other published unrelated DNM datasets, including cases of male infertility [13], autism spectrum disorder [10], schizophrenia [33, 34], undiagnosed developmental disorders [35], congenital heart disease [11], Tourette disorder [15, 31], and other unaffected individuals [10] (Fig. 4b). This suggested the specific contributions of the four candidates to female infertility with oocyte and embryo defects. Among the four genes, *TUBA4A* showed the most significant enrichment of DNMs in infertile parent–child trios ($p = 1.05 \times 10^{-6}$). There were three DNMs (E77K, p.L286P, and C347Y) in *TUBA4A* from three independent trios (Fig. 4c), and the mutation in Trio 3 was verified at the RNA level using the patient's granulosa cells (Additional file 1: Fig. S5).

The three infertile individuals with *TUBA4A* DNMs shared similar phenotypes of embryonic arrest according to their clinical information (Additional file 2: Table S6). The proband in Trio 1 was 32 years old and had been diagnosed with primary infertility for 7 years. She had undergone 3 IVF/ICSI attempts, and only 4 cleaved embryos were retrieved. The proband in Trio 2 was 29 years old. She had undergone 2 IVF/ICSI, and only 2 cleaved embryos were retrieved respectively. The proband in Trio 3 was 38 years old and she has a normal menstrual cycle and has no endocrine disorders. During her 15 years of infertility, she had undergone 8 IVF/ICSI cycles. Altogether, 68 oocytes were retrieved, of which only 3 cleaved embryos were obtained. Viable embryos were transferred into the uterus, but failed to conceive in the three

(See figure on next page.)

**Fig. 4** Identification of pathogenic mutations in *TUBA4A*. **a** Estimation of the damaging recurrent DNM genes. *P*-values and *q* values for recurrence were calculated using TADA (the de novo-only algorithm) in our cohort. Based on previously published thresholds ($q \leq 0.1$), *PCDH20* ($q = 0.42$) and *DGKZ* ($q = 0.51$) did not meet this threshold and were removed from downstream analyses. **b** Enrichment of four candidate genes compared with seven external de novo datasets, including male infertility (MI), undiagnosed developmental disorders (UDD), autism spectrum disorder (ASD), schizophrenia (SCZ), congenital heart disease (CHD), Tourette disorder (TD), and unaffected controls (CT). *x* axis was labeled as "disease_sex_sample number" and mix means mixed sample. *P*-values were calculated using the Poisson cumulative density function in external datasets. **c** Three infertile parent-child trios with de novo *TUBA4A* mutations. Squares denote male family members, circles denote female members, and solid symbols represent affected members. The " = " sign indicates infertility. **d** Nine pedigrees of sporadic *TUBA4A* mutations. Squares denote male family members, circles denote female members, and solid symbols represent affected members. The " = " sign indicates infertility

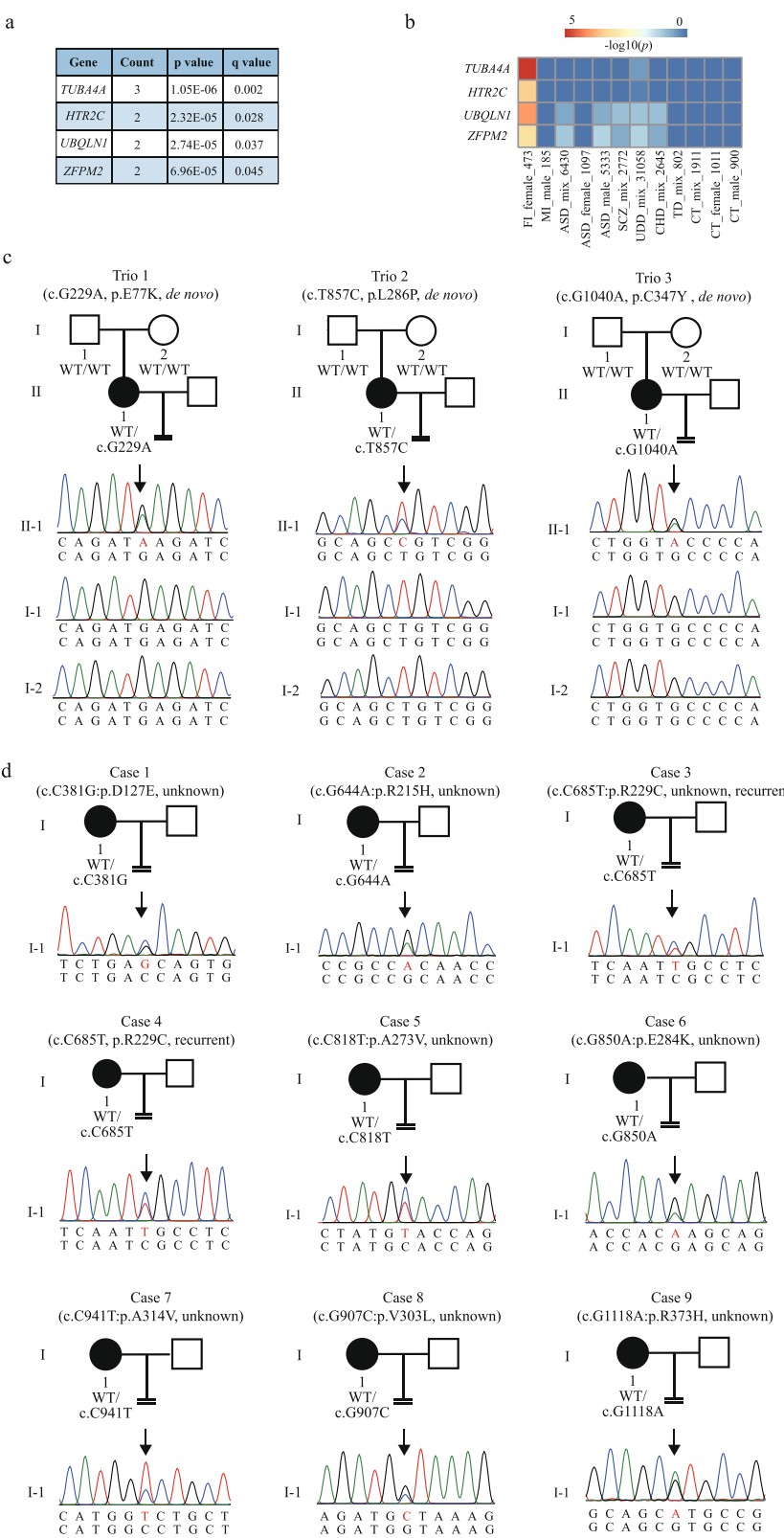

**Fig. 4** (See legend on previous page.)

affected individuals. The detailed clinical information and a full description of these individuals are provided in Additional file 2: Table S6.

To further confirm the genetic contribution of *TUBA4A* in female infertility with oocyte and embryo defects, we performed gene burden test among 1155 sporadic female infertility cases and 2813 internal controls. Similarly, *TUBA4A* was also enriched in sporadic cases without parent information ($p = 4.42 \times 10^{-4}$; Fisher's exact test). An additional nine sporadic cases with *TUBA4A* mutations were identified and verified by clinical information and Sanger sequencing (Fig. 4d; Additional file 2: Table S6, Table S7, and Table S8). Of nine sporadic cases, seven were with phenotype of early embryonic arrest and two with phenotype of oocyte maturation arrest. The above genetic evidence and clinical information strengthen the pathogenic role of *TUBA4A* mutations in female infertility with oocyte and embryo defects. Therefore, we focused on *TUBA4A* as a representative of DNM gene for performing functional assays.

### TUBA4A is evolutionarily conserved across species and is highly expressed in human oocytes

*TUBA4A* contains four exons encoding a 448 amino acid protein, and all heterozygous mutations were located in exon 3 and exon 4 (Fig. 5a). Multiple sequence alignment indicated that the amino acids that were altered in these affected individuals are highly conserved among mammalian species, suggesting the evolutionarily conserved function of TUBA4A. To determine the temporal expression of human *TUBA4A* in female germ cells, the single-cell transcriptomic dataset from human fetal FGCs was processed with the Seurat pipeline. As shown in Fig. 5b, *TUBA4A* was preferentially expressed in dictyate oocytes compared to other stages.

In addition, we compared the expression of *TUBA4A* mRNA in human oocytes, early embryos, and other somatic tissues including heart, liver, spleen, lung, kidney, brain, spine, and granulosa cells. As shown in Fig. 5c, *TUBA4A* was highly expressed in human oocytes and weakly expressed in other somatic tissues. Based on these findings, *TUBA4A* may play an important role during human oocyte maturation and early embryo development.

### TUBA4A mutations cause microtubule destabilization

To explore the potential effects of *TUBA4A* mutations, the wild-type protein structure of TUBA4A was predicted by AlphaFold2 [36]. As shown in Fig. 6a, residues of D127, R215, L286 (de novo), V303, and R373 were buried within the α-tubulin subunit, and these mutations could destabilize its folding; R229 were close to the GTP ligand, and these mutations could disrupt GTP hydrolysis; and E77 (de novo), A273, E284, A314, and C347 (de novo) were involved in chains or proteins binding, and these mutations could affect α/β dimer assembly. Taken together, mutations in *TUBA4A* may affect dimer assembly and microtubule stability.

To further determine the functional impairment of *TUBA4A* mutations on microtubule behavior in vitro, we transfected FLAG-tagged *TUBA4A* plasmids into cultured HeLa cells. In the group expressing wild-type protein, *TUBA4A* was co-assembled into a normal microtubule network (Fig. 6b) at low and intermediated expression

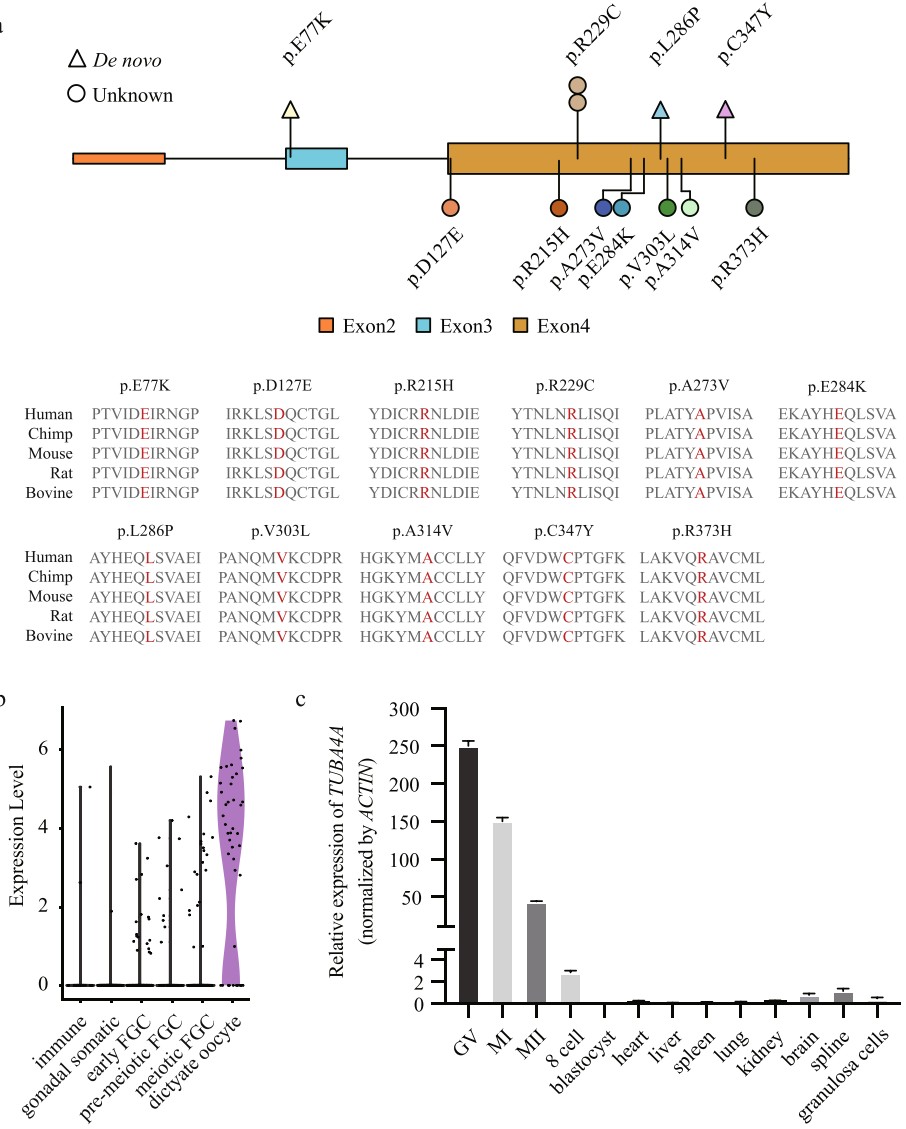

**Fig. 5** *TUBA4A* is evolutionarily conserved across species and highly expressed in human oocytes. **a** Distribution and evolutionary conservation of *TUBA4A* mutations. Mutations identified in this study are marked as colored lollipop. The conservation analysis of the mutant amino acids is indicated by the alignment of five mammalian species and mutant residues are shown in red. **b** Violin plot showing the expression dynamics of *TUBA4A* during folliculogenesis. **c** The relative expression of *TUBA4A* mRNA in different stages of human oocytes, embryos, and several somatic tissues and normalized to *ACTIN* mRNA. The bars show the mean of three separate measurements, and error bars denote standard deviations

levels, whereas high expression of TUBA4A caused an abnormal microtubule network, indicating a dosage effect on microtubule disruption. Compared with the wild-type protein, all of the DNMs (E77K, L286P, and C347Y) caused severe microtubule destabilization (Fig. 6c). As for the eight mutations from sporadic cases, six mutations (R215H, R229C, A273V, E284K, A314V, and R373H) were incorporated in microtubules with a more severely abnormal appearance compared with wild-type. The different extent of microtubule disruption might be due to the different effects of each identified mutation. Thus, we conclude that all of the DNMs and most of the sporadic

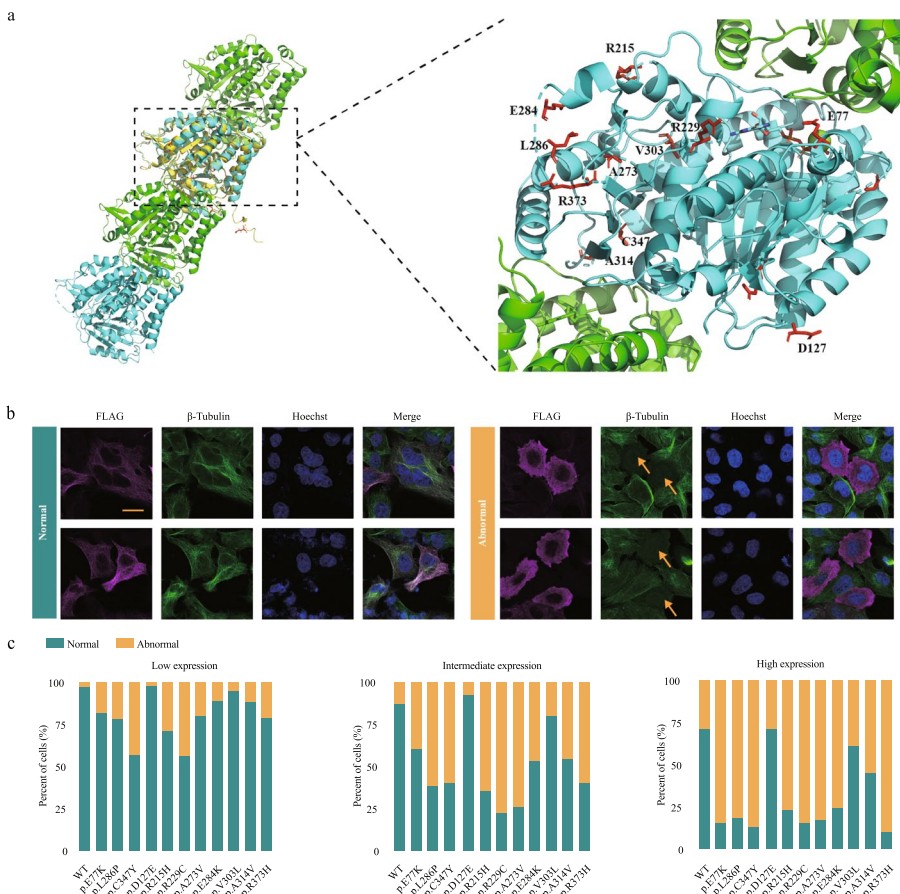

**Fig. 6** TUBA4A mutations cause microtubule destabilization. **a** Structural implications of TUBA4A amino acid substitutions. Structure of two consecutive α/ß tubulin heterodimers. Predicted wild-type structure of TUBA4A was shown in yellow and aligned to known tubulin structure (PDB code, 3JAS). Mutant residues in α tubulin are marked in red. **b** Examples of cells expressing the various FLAG-tagged TUBA4A plasmids, with microtubules assembled into a normal or abnormal interphase network. The scale bar indicates 10 μm. **c** Quantitative analysis of the microtubule phenotypes shown in **b**. Low expression of mutant TUBA4A was typically associated with normal phenotypes, whereas intermediated and high expression of mutant TUBA4A was typically associated with abnormal phenotypes. Approximately 200 transfected cells expressing either wild-type or mutant TUBA4A were evaluated in each of two separate experiments

mutations in *TUBA4A* cause significant microtubule destabilization to different extents through a dominant-negative mechanism (Fig. 6c).

## Mutant TUBA4A cRNAs result in abnormalities in mouse oocyte maturation and embryonic development

Finally, to establish the causal relationship between *TUBA4A* mutations and the corresponding phenotypes, we first evaluated the effects of *TUBA4A* mutations on oocyte maturation through microinjection of wild-type or mutant cRNAs into mouse GV oocytes. In the control group, the rate of oocyte maturation was normal (84.7% ± 2.7%). In contrast, microinjection of several *TUBA4A* mutant cRNAs (8 out of 11 mutations) significantly reduced the rate of first polar body extrusion to 24.5–66.3% (Fig. 7a). Next, we assessed the effects of *TUBA4A* mutations on embryo development through microinjection of wild-type or mutant cRNAs into mouse zygotes.

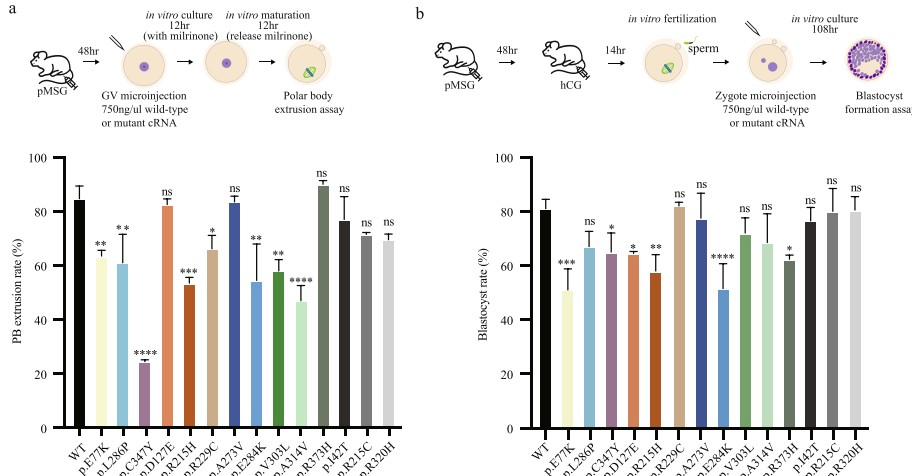

**Fig. 7** Effects of mutations on polar body extrusion and blastocyst formation in mice. **a** The effects of the mutations on polar body (PB) extrusion. The above shows the diagram of GV oocytes collection, microinjection, and polar body extrusion assay. The below shows the statistical analysis of polar body extrusion rate of wild-type and mutations. Significance was compared between wild-type and mutant groups. Two independent experiments were performed and followed Fisher's exact test. * $p < 0.05$; ** $p < 0.01$; *** $p < 0.001$; **** $p < 0.0001$; ns, not significant. **b** The effects of the mutations on blastocyst formation. The above shows the diagram of zygote collection, microinjection, and blastocyst formation assay. The below shows the statistical analysis of blastocyst rate of wild-type and mutations. Two independent experiments were performed and followed Fisher's exact test. * $p < 0.05$; ** $p < 0.01$; *** $p < 0.001$; **** $p < 0.0001$; ns, not significant

For the wild-type *TUBA4A* cRNA, 81.0% ± 2.0% of microinjected zygotes developed to the blastocyst stage. However, microinjection of several *TUBA4A* mutant cRNAs (6 out of 11 mutations) resulted in embryonic development arrest and reduced the rate of blastocyst formation to 51.0–65.0% (Fig. 7b).

To confirm the specificity of the phenotype resulting from the *TUBA4A* mutations we identified, parallel experiments were also performed in the control group, including one mutation (p.I42T) that was found in our collected in-house control database (high frequency with 0.24% in the gnomAD east Asian database) and two reported mutations (p.R215C, p.R320H) associated with familial amyotrophic lateral sclerosis (ALS) [37]. Microinjection of all three control cRNAs had no effect on mouse oocyte maturation or embryonic development (Fig. 7a, b). These results suggest that the *TUBA4A* mutations identified in our patients affect oocyte maturation or embryonic development.

## Discussion

Here, we present the first comprehensive analysis of rare DNMs in infertile females and identified a set of 481 DNMs. The corresponding DNM genes were involved in the pathways associated with female reproductive processes. Among the genes, *TUBA4A* was the most significantly enriched in our infertile parent–child trios. In vitro experiments in HeLa cells and mouse oocytes demonstrated the pathogenic role of *TUBA4A* mutations.

Although DNMs have been found to play an importance in several kinds of human diseases [10, 12, 31, 38, 39], there have been no comprehensive reports on the role of DNMs in female infertility with oocyte and embryo defects. In this study, a total of 481 DNMs were identified in our cohort. By combining single-cell gene expression data from

female FGCs, we found that these identified DNM genes were involved in the meiosis, embryonic development, and reproductive structure development. Among the 481 DNMs, about 40 genes were presumed to be involved in female reproductive biological processes, accounting for 8.32% (40/481) of the genes, while DNM genes of unaffected siblings and external control individuals had no significant association with female reproductive processes. We found 4 genes (*TUBA4A*, *UBQLN1*, *HTR2C*, and *ZFPM2*) with at least two damaging DNMs in infertile females and these 4 genes were significantly enriched in our affected individuals compared to other control groups. De novo mutations are more deleterious than inherited variants and play a role in human diseases [40]. Although de novo mutation rate is extremely low ($1.20 \times 10^{-8}$ per nucleotide per generation), de novo mutations occur with each generation, which may account for the occurrence of reproductively fatal diseases in human population [8, 40]. Based on previous published studies, we speculated that other three genes (*UBQLN1*, *HTR2C*, and *ZFPM2*) may also play a role in female infertility besides *TUBA4A* [10, 11, 31, 38, 40]. *UBQLN1* were considered a granulosa cell biomarker to predict pregnancy in ART [41]. *HTR2C* has clinically valid associations with "implantation and early development," "implantation failure," and "pregnancy loss after IVF" [42]. *ZFPM2* are required for proper development of the fetal ovary [43, 44]. In addition, we identified 468 genes with single DNM, some of which have been involved in reproductive pathways according to our GO analysis and credible database (MGI and PubMed). This suggests that the identified DNM genes play an essential role in contributing to abnormalities in oocyte maturation/embryonic development and female infertility. We believe that with increasing sample size and functional investigation, additional DNM genes associated with female infertility will be identified.

Multiple isotypes of α and β tubulin are the major polypeptide components of microtubules, which are essential for normal meiotic spindle assembly. In the present study, we found that de novo mutations in *TUBA4A*, an isotype of α tubulin, cause human oocyte and embryo defects. Mutant *TUBA4A* leads to destabilized microtubule networks and impaired oocyte maturation and embryonic development in vitro. These findings imply that TUBA4A is one of the essential subunits of α tubulin during human oocyte and embryo development. Previously, we and other groups have demonstrated that mutations in *TUBB8*, the predominant isotype of β tubulin in human oocytes, cause oocyte maturation arrest characterized by abnormal or absent meiotic spindles [2, 45–47]. These studies further indicated the essential role of α/β tubulin isotypes in normal oocyte maturation, early embryonic development, and female fertility. Mutations in the same gene may result in different disease phenotypes due to the different genetic backgrounds of each individual [48]. Firstly, Mutations in one gene may show incomplete penetrance or complete penetrance due to modifiers, epistatic interactions, or suppressors present in the genome among different individuals. Secondly, this phenomenon may be caused by other factors, such as age, sex, and environment [49–52]. Additionally, background effects were verified in other diseases caused by de novo mutations, such as congenital heart disease [11] and schizophrenia [33]. One report showed that individuals with familial ALS had *TUBA4A* mutations based on exome sequencing data from large cohorts of European and American patients [37]. However, Li and colleagues found there were no *TUBA4A* causative mutations in 580 Chinese ALS patients,

which suggested that *TUBA4A* may not be a significant genetic factor in Chinese ALS patients [53]. In this study, all affected individuals with *TUBA4A* mutations only exhibited the phenotype of female infertility without any abnormalities in somatic tissues or organs. Firstly, *TUBA4A* mutations cause different diseases which may be due to individual's background effects. Secondly, this might be due to the different thresholds of *TUBA4A* amount required between oocytes and other tissues (Fig. 5c). Finally, different mutations in *TUBA4A* may cause different phenotypes due to different pathogenicity of these mutations. In this study, mutations of *TUBA4A* cause different degrees of microtubule destabilization (Fig. 6c). Altogether, our results showed that *TUBA4A* mutations cause abnormality of oocyte maturation or embryo development.

## Conclusion

In summary, we provide genetic evidence that DNMs are strongly associated with female infertility by analyzing WES data. These findings will improve our mechanistic understanding of recurrent ART failure and provide potential genetic markers for the diagnosis of clinical patients in the future.

## Methods

### Study samples

In this study, all individuals were recruited from collaborating hospitals and reproductive centers comprising 298 infertile parent–child trios with oocyte maturation arrest, 175 infertile parent–child trios with embryonic arrest, 92 unaffected biological sibling trios, 1155 sporadic infertile females, and 2813 unaffected control females (913 controls are from our in-house database and 1900 controls are from the HuaBiao project [54]. All sample information is shown in Additional file 2: Table S1. Affected individuals were diagnosed with primary infertility for several years. They have undergone several IVF/ICSI attempts ($\geq 2$ cycles) and all failed to conceive. Two main phenotypes were observed in these patients based on their IVF/ICSI cycles: oocyte maturation arrest and early embryonic arrest. Oocyte maturation arrest can occur at the GV and MI stages, while embryonic arrest mainly refers to the failure of forming 8-cell embryos [2, 55–57]. Since the major zygotic genome activation in human was started at 8-cell embryos, therefore the embryonic arrest phenotype was mostly due to maternal factors [58–60]. This can be demonstrated by the fact that nearly all the embryo arrest-associated genes (*PADI6, NLRP2, NLRP5, BTG4, CHK1, CDC20, MEI1, FBXO43, MOS, TUBB8*) identified so far are from a maternal origin and resulted from inadequate cytoplasmic maturation of the oocyte [3, 56, 61–67]. Therefore, we combined these two cohorts in the following de novo analysis. For sample collection, a complete medical evaluation was performed: (I) age younger than 45 (early natural menopause often occurs at ages 45) [68–71]; (II) primary infertile; (III) physical examination; (IV) blood tests for hormones and infectious diseases [72, 73]; (V) the husband has normal sperm morphology and concentration; and (VI) patients had no other known causes of infertility, including chromosome anomalies, sexually transmitted infections, or surgical operations of the reproductive system. Besides, although we set the age younger than 45, the average age in our patient cohort is 31 (Additional file 1: Fig. S1d). This study was performed in accordance

with the Helsinki Declaration. The blood samples were mainly recruited from Shanghai Medical College, Fudan University and some of the hospitals following informed consent, and our study was approved by the Ethics Committee of the Biomedical Research Institute of Shanghai Medical College, Fudan University and hospitals (Ethics approval number: 162, 20,161,207, 2021–006 and JIAI E2021-33), and written informed consent was obtained from all participants.

As external controls, we collected public de novo datasets associated with other diseases, including male infertility ($N=185$) [13], schizophrenia ($N=2772$) [33, 34], Tourette disorder ($N=802$) [15, 31], congenital heart disease ($N=2645$) [11], autism spectrum disorder ($N=6430$) [10], undiagnosed development disorders ($N=31{,}058$) [35], and unaffected controls ($N=1911$) [10, 29].

### IVF/ICSI protocols
Procedure of IVF/ICSI has been described in previous studies [74, 75]. Briefly, IVF was performed in human tubal fluid (HTF; Irvine Scientific, Santa Ana, CA, USA) with 10% (v/v) serum substitute supplement (SSS; Irvine Scientific, Santa Ana, CA, USA) and ICSI was performed in separate microdroplets. The fertilization occurs within 16–20 h after IVF/ICSI treatment. All embryos were cultured in the Continuous Single Culture medium (Irvine Scientific, Santa Ana, CA, USA) at 37 °C in a humid condition with 5% $O_2$ and 6% $CO_2$. Embryos' quality was assessed and graded according to Cummins' criteria [76].

### Whole-exome sequencing and dataset quality control
Genomic DNA was extracted from peripheral blood leukocytes using the ETP-300 Nucleic Acid Extractor (Enriching), library construction and capture were processed using Twist Bioscience's Kit, and sequencing was performed on an Illumina Hiseq 3000 platform. Sequence reads were aligned to human genome build 37 (GRCh37/hg19) using the Burrows–Wheeler Aligner [77]. Sorting by chromosomes, making duplicates, and single nucleotide variant and insertion/deletion (indel) calling were performed using the Genome Analysis Toolkit (GATK, version 4.1.9.0) [78] and variants were annotated using ANNOVAR [79]. Quality Score Recalibration was used to estimate the accuracy of variant calls.

#### Filter 1: Sample-based quality control
To check the pedigree information and the relationships of the samples, relatedness was estimated by PLINK Identity-by-descent (IBD) [80]. The IBD sharing between the probands and parents in all trios was between 45 and 55%. Subsequently, the variants' metrics were generated by GATK CollectVariantCallingMetrics in all samples to remove outliers utilizing the sklern principal component analysis (PCA) module in Python.

#### Filter 2: Region-based quality control
We applied the following region-based filters across all samples: (I) exclude variants within low-complexity regions [10, 81] and (II) exclude variants within segmental duplication regions (http://humanparalogy.gs.washington.edu/build37/data/GRCh37GenomicSuperDup.tab).

### *Filter 3: Variant-based quality control*

For variant quality control, the following filters were applied: (I) exclude single nucleotide variants not passing quality score recalibration, (II) exclude calls with genotyping quality scores less than 40, (III) exclude calls with Fisher Strand value greater than 30, and (IV) exclude calls with a depth less than 10.

### Filtering and analysis of DNMs

DNMs were called in parent–child trios using in-house scripts. Candidate DNMs were filtered with the following criteria: (I) a depth of child variant site to depth of parent variant site ratio $\geq 0.3$, (II) alternative allele depth $\geq 10$, (III) exclusion of DNMs outside coding regions, and (IV) allele balance of affected probands $> 0.2$. The effects of the variants on protein function were predicted using Polymorphism Phenotyping [82] and MutationTaster [83] and were classified with ACMGG [84]. All DNMs were confirmed by in silico visualization in Integrative Genomics Viewer. Candidate DNMs were then verified by Sanger sequencing of PCR amplicons in participants who provided DNA and raw sanger sequencing data were uploaded into our GitHub code repository. In fact, there are 26 trios (13 trios with recurrent DNMs and 13 randomly selected trios) that were verified by Sanger sequencing (Fig. 4c, Additional file 1: Fig. S6).

### Burden of DNMs

The burden of DNMs in cases and unaffected siblings (controls) was processed with the denovolyzeR package [85]. We used the sequence context to derive the probability of observing a DNM in each gene as described previously [86]. Besides 92 unaffected sibling trios, a cohort of external control de novo datasets comprising 1011 unaffected females were also used [10, 29]. The overall enrichment was calculated by comparing the observed number of DNMs across each functional class to the expected number under the null mutation model using the Poisson test in R.

### Estimating the number of risk genes

The maximum likelihood estimation (MLE) procedure was established to estimate the number of risk genes via recurrent de novo events as described previously [14–16]. We defined K to be the total number of observed possibly damaging DNMs among female infertility cases. R1 indicated the number of genes mutated exactly twice in cases, and R2 indicated the number of genes mutated three times or more. The percentage of damaging DNMs carrying female infertility risk (E) was estimated with the following equation:

$$E = \frac{M1 - M2}{M1}$$

where M1 and M2 are the observed rates of damaging DNMs (probably damaging + loss-of-function (LOF)) per infertile female proband and per unaffected sibling control, respectively. We performed 25,000 permutations for every possible number of risk gene from 1 to 2500. The MLE was simulated as follows: for each permutation, we

randomly selected G risk genes from the DNM gene sets. Next, we sampled the number of contributing damaging mutations in the risk genes, C, based on the binomial (K, E) distribution. We then simulated C contributing damaging mutations in G risk genes and K-C non-contributing damaging mutations in the total gene sets using the gene's mutation rate (mu) for weighting, which accounts for gene size and GC content. In each iteration, we combined the risk genes and non-contributing genes and checked whether the recurrent mutation count was consistent with what we observed in our study (R1 and R2). We then estimated the number of risk genes using the MLE. The estimating plot was processed by the ggplot2 package in R, and we determined the MLE of risk genes to be 419 genes.

### Identification of risk genes by the transmission and de novo association test (TADA)

As a Bayesian model, TADA is useful for evaluating rare variations using whole exome sequencing data [14]. We used the TADA de novo-only model to identify risk genes. In this study, TADA de novo analysis considered missense and LOF mutations. The main input was the number of de novo missense, synonymous, and LOF mutations per gene. Additionally, the mutation rate (mu) of all de novo genes was considered in order to make the simulation more realistic. TADA analysis was processed with the following parameters:

1) Fraction of risk genes ($\pi$)

$$\pi = \frac{419 \text{ risk genes by MLE}}{17,726 \text{ refseq hg19 genes}}$$

2) Fold enrichment for missense (mis) and LOF mutations ($\lambda$)

$$For\ mis : \lambda = \frac{\text{numProbandMisMutations}}{\text{numSiblingMisMutations} * \frac{\text{numProbandSynMutations}}{\text{numSiblingSynMutations}}}$$

$$For\ LOF : \lambda = \frac{\text{numProbandLOFMutations}}{\text{numSiblingLOFVarians} * \frac{\text{numProbandSynMutations}}{\text{numSiblingSynMutations}}}$$

3) Relative risk for missense and LOF mutations ($\gamma$)

$$For\ mis : \gamma = 1 + \frac{\lambda - 1}{\pi}$$

$$For\ LOF : \gamma = 1 + \frac{\lambda - 1}{\pi}$$

With these parameters, we ran TADA de novo to calculate the Bayesian factors of all de novo genes. Next, the *p*-value of each gene was estimated by generating random mutational data based on each gene's specified mutation rate. We used 1000 samplings to obtain the null distribution. Finally, TADA de novo was used to calculate a false

discovery rate $q$-value for each gene. We considered genes with low $q$-values $\leq 0.1$ as candidate genes that are most likely pathogenic.

### Enrichment of risk genes in public databases

To determine whether risk genes were enriched in rare mutations with other genetically correlated traits, we ran a Poisson test on different DNM gene sets using R. We collected DNM gene sets from the latest research on male infertility [13], undiagnosed developmental disorders [35], autism spectrum disorder [10], schizophrenia [33, 34], congenital heart disease [11], and Tourette disorder [15, 31]. In addition, we also collected DNM genes of unaffected subjects (controls) from other public databases [10, 29].

### Processing of single-cell RNA-Seq data

Normalized expression tables of female germ cell RNA-seq datasets were collected from Li et al. (human fetal female germ cells) [19], Zhang et al. (human oocytes during folliculogenesis) [87], Takeuchi et al. (human matured oocytes) [22], and Xue et al. (human early embryos) [21]. The single-cell transcriptomic dataset of fetal female germ cells (FGC) was reanalyzed due to unclear cell development states. For all sequenced female FGCs, we counted the number of genes detected in each cell. Briefly, cells with fewer than 2000 genes or 100,000 transcripts were filtered out, and somatic cells were excluded. In total, 948 cells passed the quality control, and the R package Seurat (v4.0.2) [88] was used to analyze the single-cell data.

To analyze cell development states, $t$-distributed stochastic neighbor embedding (t-SNE) was performed using the function "RunTSNE," and the PCA plot was visualized using the function "FeaturePlot" in the Seurat workflow based on marker genes. Finally, outliers were removed based on lower and upper quantile values and mean values were calculated. (The code is available from our GitHub repository.)

### Clustering of DNM genes in germ cell lifecycle

To quantify the contribution of DNM genes in germ cell development, normalized expression values were extracted to perform a time-series clustering (early FGCs, pre-meiotic FGCs, meiotic FGCs, dictyate oocytes, primordial follicles, primary follicles, secondary follicles, antral follicles, preovulatory follicles, GV, MI, MII, zygotes, two-cell embryos, four-cell embryos, eight-cell embryos, and morula). The time-series soft clustering analysis was conducted with the fuzzy c-means algorithm in the R package Mfuzz (v2.50.0) [89] to identify different expression patterns across development stages. DNM genes of different expression clusters were used to perform separate GO analyses using DAVID (https://david.ncifcrf.gov/) [90] and the R package clusterProfiler (v4.1.4) [91]. The schematic diagram was adapted from Cordeiro et al. [92].

## Structure prediction of the TUBA4A protein

To visualize the effects of altered residues on protein structure, we obtained the structure prediction of the TUBA4A protein from AlphaFold2 [36]. The AlphaFold2 source code was retrieved from the official GitHub repository (https://github.com/deepmind/alphafold) and the requisite databases were retrieved with the included script. The predicted structures of TUBA4A are available from GitHub (https://github.com/QunATCG/DenovoMut). In addition, we mapped the mutant TUBA4A residues onto the atomic structure of tubulin (Protein Data Bank code 3JAS) [93], and the PyMOL Viewer software (https://github.com/schrodinger/pymol-open-source) was used to visualize the mutant residues. Prediction of protein structure was supported by the Medical Research Data Center of Fudan University, and protein features of tubulin were obtained from the Protein Data Bank (https://www.rcsb.org/3d-sequence/3JAS?assemblyId=1).

## Mouse oocyte/zygote collection, microinjection, and cultivation

ICR mice (Beijing Vital River Laboratory Animal Technology Co) at 7–8 weeks of age were maintained and euthanized according to procedures approved by the experimental animal ethics committee of Fudan Medical College. For evaluating the effects of *TUBA4A* (NM_006000.3) mutations on oocyte maturation, 5 IU of pregnant mare's serum gonadotropin was injected and germinal vesicle (GV) oocytes were collected from the mouse ovaries 48 h later. Approximately 5 pl of the wild-type or mutant *TUBA4A* cRNAs (750 ng/µl) were injected per oocyte. The injected GV oocytes were cultured in M16 (Sigma-Aldrich) with 2.5 µM milrinone (Sigma-Aldrich) to prevent GV breakdown. After 12-h inhibition, the GV oocytes were released into milrinone-free medium, and the polar body extrusion rate was recorded 12 h after being released. To detect the effects of mutations on embryonic development, 7.5 IU of human chorionic gonadotropin was administered 48 h after injection of pregnant mare's serum gonadotropin. MII oocytes were collected for in vitro fertilization, and zygotes were picked up for the subsequent microinjection procedure. Approximately 5 pl of the wild-type or mutant *TUBA4A* cRNAs (750 ng/µl) were injected per zygote. The injected zygotes were transferred to KSOM medium (Nanjing Aibei Biotechnology) for in vitro culture, and the embryos were assessed at 108 h after fertilization to determine the blastocyst formation rate.

## Cell culture and transfection

HeLa cells were cultured in Dulbecco's modified Eagle medium supplemented with 10% fetal bovine serum and 1% penicillin/streptomycin (Gibco) and maintained in a humidified incubator at 37 °C with 5% $CO_2$. Wild-type and mutant plasmids of *TUBA4A* were transfected into HeLa cells using the PolyJet In Vitro DNA Transfection Reagent (SignaGen) according to the manufacturer's instructions. After 36-h culture, the transfected cells were harvested for the following cell immunostaining procedures.

### Cell immunostaining

To evaluate the effects of the identified mutations on microtubule behavior, the transfected HeLa cells were fixed in 4% paraformaldehyde in PBS and blocked at room temperature for 1 h. Anti-FLAG-Cy3 antibody (1:500 dilution, Sigma-Aldrich) was used for determining the TUBA4A-FLAG localization, anti-β-tubulin antibody (1:500 dilution, Sigma-Aldrich) was used for detecting the endogenous microtubule network, and the DNA was labeled with Hoechst 33,342 solution (1:700 dilution, BD). The HeLa cells were observed and images were captured on a confocal laser-scanning microscope (LSM880, Zeiss) with a $63 \times$ oil objective. For the quantification of microtubule phenotypes, approximately 200 cells expressing either wild-type or mutant TUBA4A were detected and classified according to the level of expression of the overexpressed plasmids (as judged by the intensity of the fluorescent label) in each of two independent experiments.

## Supplementary informationSupplementary Information

---

**Additional file 1.** Supplementary Figures.

**Additional file 2: Table S1.** The clinical characteristics of individuals used in our cohort. **Table S2.** DNMs identified in this study. **Table S3.** GO terms of probands and unaffected individuals. **Table S4.** Selected genes and GO terms involved in female reproduction. **Table S5.** Genes associated with female reproduction defects. **Table S6.** Clinical IVF/ICSI information of patients with *TUBA4A* mutations. **Table S7.** *TUBA4A* pathogenic heterozygous mutations observed in the 12 families. **Table S8.** Primers used in this study.

**Additional file 3.** Review history.

---

#### Acknowledgements
We thank Prof. Jiucun Wang for providing control samples from HuaBiao project.

#### Peer review information

#### Review history
The review history is available as Additional file 3.

#### Authors' contributions
The project was conceived and supervised by QS, LW, and QL. QL performed plasmid constructions and verification-related experiments with the help of LZ, YZ, HZF, HG, and QLL. YPK, YCG, SRX, BT, LW, XYM, XXS, JZS, PX, FYD, SGX, SHB, QXM, PY, WJW, NM, DS, and BX performed fertility assessment or provide samples. JD, JM, and ZHZ provided advice on this study. HZF and HG extracted genomic DNA. LH, LJ, LW, and QS guided the development and evaluation. QL and BBC performed the bioinformatics analyses and designed the experiments. QL and LZ wrote the manuscript with help from all the authors. QS and LW participated in the drafting or revising of the manuscript. The authors read and approved the final manuscript.

#### Funding
This work was supported by the National Natural Science Foundation of China (82288102), the National Key Research and Development Program of China (2021YFC2700100), the National Natural Science Foundation of China (32130029, 81725006, 82171643, 81971450, 81971382, 82071642, and 82001538), the Project of the Shanghai Municipal Science and Technology Commission (21XD1420300 and 19JC1411001), the Natural Science Foundation of Shanghai (19ZR1444500 and 21ZR1404800), the Capacity Building Planning Program for Shanghai Women and Children's Health Service, the collaborative innovation center project construction for Shanghai Women and Children's Health, and the Guangdong Science and Technology Department Guangdong/Hong Kong/Macao Joint Innovation Project (2020A0505140003).

#### Availability of data and materials
The main data relevant to the study are included in the article or uploaded as supplementary information. The scripts of the main steps and raw Sanger sequencing data can be found in the GitHub (https://github.com/QunATCG/DenovoMut) [94]. The variation data for patients reported in this paper have been deposited in the Genome Variation Map (GVM) [95] in National Genomics Data Center, Beijing Institute of Genomics, Chinese Academy of Sciences, and China National Center for Bioinformation, under accession number GVM000497 [96]. Variants of control cohort used in this study were generated by HuaBiao project and can be obtained from https://www.biosino.org/wepd/.

## Declarations

### Ethics approval and consent to participate

Our study was approved by the Ethics Committee of the Biomedical Research Institute of Shanghai Medical College, Fudan University and hospitals (Ethics approval number: 162, 20161207, 2021–006 and JIAI E2021-33), and written informed consent was obtained from all participants.

### Consent for publication

Not applicable.

### Competing interests

The authors declare that they have no competing interests.

### Author details

[1]Institute of Pediatrics, Children's Hospital of Fudan University, the Shanghai Key Laboratory of Medical Epigenetics, the Institutes of Biomedical Sciences, the State Key Laboratory of Genetic Engineering, Fudan University, Shanghai 200032, China. [2]Human Phenome Institute, Fudan University, Shanghai 200438, China. [3]Reproductive Medicine Center, Shanghai Ninth Hospital, Shanghai Jiao Tong University, Shanghai 200011, China. [4]Department of Reproductive Medicine, The Third Affiliated Hospital of Zhengzhou University, Zhengzhou 450052, China. [5]NHC Key Lab of Reproduction Regulation (Shanghai Institute for Biomedical and Pharmaceutical Technologies), Fudan University, Shanghai 200032, China. [6]Fertility Center, Shenzhen Zhongshan Urology Hospital, Shenzhen 518001, Guangdong, China. [7]Reproductive Medicine Center, The First People's Hospital of Changde City, Changde 415000, China. [8]Shanghai Ji Ai Genetics and IVF Institute, Obstetrics and Gynecology Hospital, Fudan University, Shanghai 200011, China. [9]Reproductive Medicine Center, Northwest Women's and Children's Hospital, Xi'an 710000, China. [10]Hainan Jinghua Hejing Hospital for Reproductive Medicine, Haikou 570125, China. [11]Reproductive Medicine Center, Jiangsu Province Hospital, Nanjing 210036, China. [12]Reproductive Medicine Center, School of Medicine, Shanghai East Hospital, Tongji University, Shanghai, China. [13]Department of Reproductive Immunology, Shanghai First Maternity and Infant Hospital, Tongji University School of Medicine, Shanghai 201204, China. [14]Center for Reproduction and Genetics, The Affiliated Suzhou Hospital of Nanjing Medical University, Suzhou 215000, China. [15]IVF Center, Department of Obstetrics and Gynecology, Sun Yat-Sen Memorial Hospital, Sun Yat-Sen University, Guangzhou 510120, China. [16]Reproductive Medical Center, Maternal and Child Health Care Hospital of Hainan Province, Haikou 570206, Hainan Province, China. [17]Naval Medical University, Changhai Hospital, Shanghai, China. [18]Reproductive Medicine Centre, Tongji Hospital, Tongji Medical College of Huazhong University of Science and Technology, Wuhan 430030, China. [19]Bio-X Center, Key Laboratory for the Genetics of Developmental and Neuropsychiatric Disorders, Ministry of Education, Shanghai Jiao Tong University, Shanghai 200030, China. [20]State Key Laboratory of Genetic Engineering and Collaborative Innovation Center for Genetics and Development, School of Life Sciences, Fudan University, Shanghai 200438, China.

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

## 