## [**Additional file 3.** Review history. · Genome Biology]

Review History

First round of review

Reviewer 1

Are you able to assess all statistics in the manuscript, including the appropriateness of statistical tests used? Yes, and I have assessed the statistics in my report.

Comments to author:

Li et al. describe a study into the role of de novo mutations in female infertility. As there is very little information about this, it is in itself a very important study. Unfortunately, the study lacks essential information to properly evaluate this work. First and most important of all, the cohorts used in this study are not described in any detail to allow for an evaluation of the quality of the work. Secondly, the paper is very imbalanced, with more than half of the paper discussing a single gene, TUBA4A. While of interest, that leaves very little details on the de novo work which is the title and the focus of the paper.

1. Patient cohort.

It is essential to know details about the patients included in this study, with sufficient detail about the clinical definitions, as well as inclusion and exclusion criteria. In the results the authors refer to '...parent-child trios with recurrent failure of IVF/ICSI attempts due to oocyte maturation arrest, embryo arrest, or recurrent miscarriage.' In the Methods part, study samples, nothing is mentioned about IVF/ICSI procedure, and no details are provided for individual patients, not even a good Table with clinical features is presented. In addition, the study is not describing a single cohort, as mentioned above it represents a combination of 3 different clinical cohorts, for which the female factor is not equally clear. For the first cohort, females were diagnosed with oocyte maturation arrest, a clear infertility phenotype. The two other cohorts are less clearly linked to a female factor: Cohort two is a cohort where there is embryo arrest, and cohort three is a cohort of women with recurrent miscarriage. No further detail is presented to show further that this is all likely due to female infertility, in fact recurrent miscarriage is often seen as very distinct from female infertility. It is very common (1-2% of women) and associated with advanced maternal age. The fact that women were included up to 45 years of age does not indicate any selection for a genetic factor. Similarly, the replication cohort consists of 1,219 sporadic infertile females without any further detail provided. It is really difficult to combine these very different cohorts in one paper, but that is unfortunately what the authors did. There is very little value in the current analysis of DNMs in this combined cohort. In addition, the authors mention that the study was approved by the Ethics Committee of Fudan University. It is unclear to me if all patients were recruited from this University as there are at least 17 hospitals mentioned as co-authors on the paper.

2. TUBA4A gene

Half way the results section the authors focus on TUBA4A (as an aside, they state they focused on nine genes but don't discuss the other 8 genes). The work on de novo mutations in TUBA4A is in itself interesting and potentially relevant for female infertility. Of interest, the authors here describe their cohort as 'abnormalities in oocyte and early embryonic development', which probably is a better description than female infertility. Unfortunately, the work suffers again from a lack of clinical information as a first point to discuss is what the detailed phenotype was

for the three women with de novo mutations in this gene (as well as the women from the sporadic replication cohort with mutations in this gene), nothing is mentioned about this at all. The rest of the work on this gene is of interest and worth publishing, if supported by good clinical information. One further remark about the Sanger sequencing images in Figure 4, they are really not that clear and not always very convincingly showing validation of mutations in both the trios (e.g. trio 3 is not very clear) as well as the sporadic cases...

3. Do de novo mutations are strongly associated with female infertility?

Unfortunately, the authors do not present convincing data to support their conclusion that de novo mutation are strongly associated with female infertility. It is unclear on what this conclusion is based, as the statistical analyses presented show an enrichment of LoF mutations (representing 12% of all DNMs), and some GO analyses that show DNMs in genes involved in 'pathways involved in female reproduction'. For this last analysis, it is unclear whether these DNMs were predicted to be pathogenic, which is critical as most DNMs will be benign. Again, as mentioned it would be good to present much more analyses for the three separate cohorts to convincingly show that de novo mutations are indeed strongly associated with female infertility.

Reviewer 2

Are you able to assess all statistics in the manuscript, including the appropriateness of statistical tests used? Yes, and I have assessed the statistics in my report.

Comments to author:

The paper is of broad interest to others in the field

Li et al present the first comprehensive analysis of rare de novo mutations (DNMs) detected by WES in infertile females and identified a set of 568 DNMs. The data suggested that the corresponding DNM genes were involved in the pathways associated with female reproductive processes. They also selected 9 genes for further evaluation, which included TUBA4A, UBQLN1, HTR2C, FRMD1, ZFPM2, ELFN2, UHRFBP1, ROBO2 and YLPM1. The data suggested that they were enriched in cohort with abnormalities in oocyte and early embryonic development compared to other published unrelated DNM datasets. The finding is novel and interesting. Further functional study illustrated that TUBA4A was highly expressed in human oocytes and weakly expressed in other somatic tissues. Mutant TUBA4A in mouse study demonstrated that TUBA4A may play an important role during human oocyte maturation and early embryo development. There are several issues need to be clarify:

Major points:

1. DNMs are known to play a prominent role in human diseases, and the utilization of trio-based exome sequencing is a sound and reliable approach. What is the meaning of Parent-child trios here? I assume the Trios should be infertile females and their unaffected parents.
2. Based on the method described, the criteria to filter the DNMs does it also include (MAF < 0.1%)? How many of the Candidate DNMs were verified by Sanger sequencing any possibility of low level mosaicism among the parents?
3. Table S1, I believe the Trio cohort is precious, for the recurrent miscarriage trios, what is the selection criteria? It seems that the DNMs are more towards oocyte maturation and early embryo

development.

4. Most of the known genetic findings towards TUBA4A is related to amyotrophic lateral sclerosis (ALS). It is very nice that the team also include the two reported mutations (p.R215C, p.R320H) associated with familial (ALS) to illustrate the differential function. Can they elaborate more regarding the possible mechanism of one gene contribute to different diseases in the discussion? Will this also explain why the rate of polar body extrusion and blastocyst formation are different between mutations?

Minor:

1. There are new single-cell profiling support TUBA4A transcriptomic changes during in vitro maturation of human oocytes <https://doi.org/10.1002/rmb2.12464>
2. Within the DNMs how many are AD, AR or X-linked?
3. Ethics approval no ref. number was provided.

Responses to Reviewer #1:

Reviewer #1: Li et al. describe a study into the role of de novo mutations in female infertility. As there is very little information about this, it is in itself a very important study. Unfortunately, the study lacks essential information to properly evaluate this work. First and most important of all, the cohorts used in this study are not described in any detail to allow for an evaluation of the quality of the work. Secondly, the paper is very imbalanced, with more than half of the paper discussing a single gene, TUBA4A. While of interest, that leaves very little details on the de novo work which is the title and the focus of the paper.

Response: *Thanks for your comments and we reanalyzed our data according to your constructive suggestions, which we believe will greatly improve the quality of our manuscript. We removed patient cohort with recurrent miscarriage and added detailed clinical information of the patients. Secondly, we added new figures and discussion to show the association between other candidate genes and female infertility.*

1. Patient cohort.

It is essential to know details about the patients included in this study, with sufficient detail about the clinical definitions, as well as inclusion and exclusion criteria. In the results the authors refer to '...parent-child trios with recurrent failure of IVF/ICSI attempts due to oocyte maturation arrest, embryo arrest, or recurrent miscarriage.' In the Methods part, study samples, nothing is mentioned about IVF/ICSI procedure, and no details are provided for individual patients, not even a good Table with clinical features is presented. In addition, the study is not describing a single cohort, as mentioned above it represents a combination of 3 different clinical cohorts, for which the female factor is not equally clear. For the first cohort, females were diagnosed with oocyte maturation arrest, a clear infertility phenotype. The two other cohorts are less clearly linked to a female factor: Cohort two is a cohort where there is embryo arrest, and cohort three is a cohort of women with recurrent miscarriage. No further detail is presented to show further that this is all likely due to female infertility, in fact recurrent miscarriage is often seen as very distinct from female infertility. It is very common (1-2% of women) and associated with advanced maternal age. The fact that women were included up to 45 years of age does not indicate any selection for a genetic factor. Similarly, the replication cohort consists of 1,219 sporadic infertile females without any further detail provided. It is really difficult to combine these very different cohorts in one paper, but that is unfortunately what the authors did. There is very little value in the current analysis of DNMs in this combined cohort. In addition, the authors mention that the study was approved by the Ethics Committee of Fudan University. It is unclear to me if all patients were recruited from this University as there are at least 17 hospitals mentioned as co-authors on the paper.

Response: *We are sorry for the confusion. We agree that recurrent miscarriage is distinct from female infertility. So we have removed the cohort of women with recurrent miscarriage and reanalyzed our data in revised manuscript. What's more, we add more clinical information about our cohort, and we checked cohorts in present study carefully. Clinical information of all affected probands with TUBA4A DNMs were showed in Additional file 2: Table S6.*

Procedure of IVF/ICSI have been described in previous studies^{1,2}. Briefly, IVF was performed in human tubal fluid (HTF; Irvine Scientific, Santa Ana, CA, USA) with 10% (v/v) serum substitute supplement (SSS; Irvine Scientific, Santa Ana, CA, USA) and ICSI was performed in separate microdroplets. The fertilization occurs within 16-20 hours after IVF/ICSI treatment. All embryos were cultured in the Continuous Single Culture medium (Irvine Scientific, Santa Ana, CA, USA) at 37 °C in a humid condition with 5% O₂ and 6% CO₂. Embryos' quality was assessed and graded according to the Cummins' criteria³.

Female infertility can be resulted from several reasons, including but not limited to, oocyte maturation arrest, premature ovarian insufficiency (POI), fertilization failure and early embryonic arrest⁴. In our present study, all affected individuals were diagnosed with primary infertility for several years. They have undergone several IVF/ICSI attempts (≥ 2 cycles) and all failed to conceive. Two main phenotypes were observed in these patients based on their IVF/ICSI cycles: oocyte maturation arrest and early embryonic arrest. Oocyte maturation arrest can occur at the GV stage and MI stage, while embryonic arrest mainly refers to the failure of forming 8-cell embryos. Since the major zygotic genome activation in human was started at 8-cell embryos, therefore the embryonic arrest phenotype was mostly due to maternal factors⁵⁻⁷. This can be demonstrated by the fact that nearly all the embryo arrest associated genes (PADI6, NLRP2, NLRP5, BTG4, CHK1, CDC20, MEI1, FBXO43, MOS, TUBB8) identified so far are from a maternal origin and resulted from inadequate cytoplasmic maturation of the oocyte⁸⁻¹⁶. So, we believe that oocyte maturation arrest and embryo arrest are most likely due to oocyte maturation defects and these two cohorts can be combined. Besides, several published de novo mutations studies integrate multiple phenotype data to identify risk genes, such as male infertility (azoospermia and oligozoospermia)¹⁷, developmental disorders (data were from three centers and patients were selected for inclusion in this study based on having one or more HPO phenotypes, such as intellectual disability, epilepsy...)¹⁸

Recurrent miscarriages have multiple etiologies, including chromosomal abnormalities, immune dysfunction, and various endocrine disorders. More than 90% were found to have at least one chromosomal abnormality in one or more cells in all blastomeres in preimplantation human embryos¹⁹. In our study, probands with recurrent miscarriages were diagnosed with infertility without known causes. So, we previously put these data into our analysis. We agree with your concern that the causes of recurrent miscarriages are complex, and the cohort is difficult to combine with oocyte maturation arrest and early embryonic arrest. Therefore, we removed the cohort of recurrent miscarriages in the revised manuscript.

Finally, to ensure that the infertility is caused by the female factor, the inclusion criteria were as follows: (1) Age younger than 45 (early natural menopause often occurs at age 45)²⁰⁻²²; (2) Primary infertility; (3) Pass physical examination; (4) Blood tests for hormones and infectious diseases^{23,24}; (5) The husband has normal sperm morphology and concentration; (6) Patients had no other known causes of infertility, including chromosome anomalies, sexually transmitted infections, or surgical operations of the reproductive system. Besides, although we set the age younger than 45, the average age in our patient cohort is 31 (new added Additional file 1: Fig. S1d).

Individuals were diagnosed as primary infertility in different hospitals. However, the blood samples were mainly recruited from Shanghai Medical college, Fudan University and some of the hospitals following informed consent and approval from the Ethics Committee. We have added the statements and approval number in the method.

Now the content has been revised as follows:

In the Methods part (Study samples):

“In this study, all individuals were recruited from collaborating hospitals and reproductive centers in comprising 298 infertile parent-child trios with oocyte maturation arrest, 175 infertile parent-child trios with embryonic arrest, 92 unaffected biological sibling trios, 1,155 sporadic infertile females and 2,813 unaffected females. All samples information is shown in Additional file 2: Table S1. Affected individuals were diagnosed with primary infertility for several years. They have undergone several IVF/ICSI attempts (≥ 2 cycles) and all failed to conceive. Two main phenotypes were observed in these patients based on their IVF/ICSI cycles: oocyte maturation arrest and early embryonic arrest. Oocyte maturation arrest can occur at the GV and MI stage, while embryonic arrest mainly refers to failure of forming 8-cell embryos [2, 50-52]. Since the major zygotic genome activation in human was started at 8-cell embryos, therefore the embryonic arrest phenotype was mostly due to maternal factors [53-55]. This can be demonstrated by the fact that nearly all the embryo arrest associated genes (PADI6, NLRP2, NLRP5, BTG4, CHK1, CDC20, MEI1, FBXO43, MOS, TUBB8) identified so far are from a maternal origin and resulted from inadequate cytoplasmic maturation of the oocyte [3, 51, 56-62]. Therefore, we combined these two cohorts in the following de novo analysis. For sample collection, a complete medical evaluation was performed: (I) Age younger than 45 (early natural menopause often occurs at ages 45) [63-66]; (II) Primary infertile; (III) Physical examination; (IV) Blood tests for hormones and infectious diseases [67, 68]; (V) The husband has normal sperm morphology and concentration; (VI) Patients had no other known causes of infertility, including chromosome anomalies, sexually transmitted infections, or surgical operations of the reproductive system. Besides, although we set the age younger than 45, the average age in our patient cohort is 31 (Additional file 1: Fig. S1d). The blood samples were mainly recruited from Shanghai Medical college, Fudan University and some of the hospitals following informed consent and our study was approved by the Ethics Committee of the Biomedical Research Institute of Shanghai Medical college, Fudan University and hospitals Fudan University (Ethics approval number: 162, 20161207, 2021-006 and JIAI E2021-33), and written informed consent was obtained from all participants.”

In the Methods part (IVF/ICSI protocols):

“Procedure of IVF/ICSI have been described in previous studies [69, 70]. Briefly, IVF was performed in human tubal fluid (HTF; Irvine Scientific, Santa Ana, CA, USA) with 10% (v/v) serum substitute supplement (SSS; Irvine Scientific, Santa Ana, CA, USA) and ICSI was performed in separate microdroplets. The fertilization occurs within 16-20 hours after IVF/ICSI treatment. All embryos were cultured in the Continuous Single Culture medium (Irvine Scientific, Santa Ana, CA, USA) at 37 °C in a humid condition with 5% O₂ and 6% CO₂. Embryos' quality was assessed and graded according to the Cummins' criteria [71]”

2. TUBA4A gene

Half way the results section the authors focus on TUBA4A (as an aside, they state they focused on nine genes but don't discuss the other 8 genes). The work on de novo mutations in TUBA4A is in itself interesting and potentially relevant for female infertility. Of interest, the authors here describe their cohort as 'abnormalities in oocyte and early embryonic development', which probably is a better description than female infertility. Unfortunately, the work suffers again from a lack of clinical information as a first point to discuss is what the detailed phenotype was for the three women with de novo mutations in this gene (as well as the women from the sporadic replication cohort with mutations in this gene), nothing is mentioned about this at all. The rest of the work on this gene is of interest and worth publishing, if supported by good clinical information. One further remark about the Sanger sequencing images in Figure 4, they are really not that clear and not always very convincingly showing validation of mutations in both the trios (e.g. trio 3 is not very clear) as well as the sporadic cases...

Response: *We appreciate for the professional suggestions. We have removed cohort with recurrent miscarriage and reanalyzed the data. To identify risk genes for infertile women with oocyte or embryo defects, we performed TADA test on all DNMs and four genes with at least two damaging DNMs (TUBA4A, UBQLN1, HTR2C and ZFPM2) passed test based on previous publication's threshold²⁵. Also, we added clinical information of probands with TUBA4A mutations in this revised manuscript (new added Additional file 2: Table S6).*

De novo mutations have an important role in human diseases and the average de novo mutation rate is 1.20×10^{-8} per nucleotide per generation^{26,27}. Remarkably, DNMs are exposed to less selective pressure and are associated with human evolution and diseases^{26,28-30}. We speculate these four genes' contribution in female infertility based on following reasons: (1) In this study, these four genes (TUBA4A, UBQLN1, HTR2C and ZFPM2) showed extremely enrichment in our infertile females compared to other seven public de novo databases. What's more, seven public databases have very large numbers of trios range from 185 to 31058. This result showed that these four genes have higher mutation frequency in our cohort compared with the other seven control cohorts. Therefore, we speculate that these four genes may play a role in infertility; (2) UBQLN1 were considered to a granulosa cell biomarkers to predict pregnancy in ART³¹. HTR2C has clinically valid associations with "implantation and early development", "implantation failure" and "pregnancy loss after IVF"³². ZFPM2 are required for proper development of the fetal ovary^{33,34}. (3) There were lots of work revealed the contribution of DNMs by hypothesis testing without functional assays and our analysis pipeline were widely used in DNM studies^{25,35,36}. Based on above reasons, we speculated that these three genes were associated with female infertility. Also, we added discussion about these three genes in the discussion part. To establish the causal relationship between DNMs and oocyte or embryo defects, we performed functional assays on the effects of DNMs in a representative gene TUBA4A, which showed the most significant enrichment of DNMs in the patient trios. We believe that with increasing sample size and functional investigation, additional DNM genes associated with female infertility will be identified. We added clinical information of probands with TUBA4A mutations (new added Additional file 2: Table S6) and these affected probands with TUBA4A mutations shared similar phenotype of embryo arrest.

We re-illustrate the sanger sequencing results in a screenshot from Chromas software (<http://technelysium.com.au/wp/chromas/>) (Fig. 4c and Fig. 4d). All trios and affected individuals

with TUBA4A mutations were verified by sanger sequencing and the resolution of all sanger sequencing images have been highly improved. What's more, the mutation in Trio 3 was verified at the RNA level using the patient's granulosa cells (Additional file 1: Fig. S5). In fact, there are 26 trios (13 recurrent trios and 13 randomly selected trios) were verified by sanger sequencing (new added Additional file 1: Fig. S6). To make our results convincing, all raw sanger sequencing data were upload into our GitHub code repository.

(<https://github.com/QunATCG/DenovoMut/tree/main/sangerSequencing>)

Also, we agree that after removing the recurrent miscarriage cohort, "abnormalities in oocyte and early embryonic development" is a better description than female infertility. So, we modified the description of female infertility to "female infertility with oocyte and embryo defects". Also, we changed our title to "Large-scale analysis of de novo mutations identifies risk genes for female infertility characterized by oocyte and early embryo defects"

Now the content has been revised as follows:

In the result part (TUBA4A showed the most enrichment of mutations in our cohort):

"The three infertile individuals with TUBA4A DNMs shared similar phenotypes of abnormalities in embryo development according to their clinical information (new added Addition file 2: Table S6). The proband in Trio 1 was 32 years old and had been diagnosed with primary infertility for 7 years. She had undergone 3 IVF/ICSI attempts, and only 2, 1 and 1 cleaved embryos were retrieved respectively. The proband in Trio 2 was 29 years old. She had undergone 2 IVF/ICSI, and only 2 cleaved embryos were retrieved respectively. The proband in Trio 3 was 38 years old and she has a normal menstrual cycle and has no endocrine disorders. During her 15 years of infertility, she had undergone 8 IVF/ICSI cycles. Altogether, 68 oocytes were retrieved, of which only 3 cleaved embryos were obtained. Viable embryos were transferred into the uterus but failed to conceive among these three affected individuals. The detailed clinical information and a full description of these individuals are provided in Additional file 2: Table S6."

In the Discussion part:

"We found 4 genes (TUBA4A, UBQLN1, HTR2C, and ZFPM2) with at least two damaging DNMs in infertile females and these 4 genes were significantly enriched in our affected individuals compared to other control groups. De novo mutations are more deleterious than inherited variants and play a role in human diseases [41]. Although de novo mutation rate is extremely low (1.20×10^{-8} per nucleotide per generation), de novo mutations occur with each generation, which may account for occurrence of reproductively fatal diseases in human population [41, 42]. Based on previous published studies, we speculated that other three genes (UBQLN1, HTR2C and ZFPM2) may also play a role in female infertility besides TUBA4A [10, 11, 32, 41, 43]. UBQLN1 were considered to a granulosa cell biomarker to predict pregnancy in ART [44]. HTR2C has clinically valid associations with "implantation and early development", "implantation failure" and "pregnancy loss after IVF" [45]. ZFPM2 are required for proper development of the fetal ovary [46, 47]. In addition, we identified 468 genes with single DNM, some of which have been involved in reproductive pathways according to our GO analysis and credible database (MGI and PubMed). This suggests that the identified DNM genes play an essential role in contributing to abnormalities

in oocyte maturation/embryonic development and female infertility. We believe that with increasing sample size and functional investigation, additional DNM genes associated with female infertility will be identified.”

In the Methods part (Filtering and analysis of DNMs):

“Candidate DNMs were then verified by Sanger sequencing of PCR amplicons in participants who provided DNA and raw sanger sequencing data were upload into our GitHub code repository”

3. Do de novo mutations are strongly associated with female infertility?

Unfortunately, the authors do not present convincing data to support their conclusion that de novo mutation are strongly associated with female infertility. It is unclear on what this conclusion is based, as the statistical analyses presented show an enrichment of LoF mutations (representing 12% of all DNMs), and some GO analyses that show DNMs in genes involved in 'pathways involved in female reproduction'. For this last analysis, it is unclear whether these DNMs were predicted to be pathogenic, which is critical as most DNMs will be benign. Again, as mentioned it would be good to present much more analyses for the three separate cohorts to convincingly show that de novo mutations are indeed strongly associated with female infertility.

Response: *By removing recurrent miscarriage patients, stringently correcting for multiple statistical tests and curating these DNM genes in MGI/PubMed database, we found DNMs is really strongly associated with female infertility characterized by oocyte maturation arrest and early embryo arrest.*

Our conclusions are based on following aspects: (1) It is well known that LoF mutations may cause severe phenotypes in human diseases³⁷⁻³⁹ and this method (enrichment of LoF mutations) were widely used in other studies^{40,41}. Our data showed specifically significant enrichment of LoF mutations in female infertility cases, compared to two independent controls, unaffected siblings and public controls (enrichment = 1.47; $p = 2.5 \times 10^{-3}$; Fig. 2C). What's more, many LoF mutations we identified have been shown to be associated with human infertility^{4,42}; (2) GO analysis is an established guideline in analyzing association relationship between DNMs and human diseases^{35,43,44}. GO analyses in our cohort showed that DNM genes were strongly involved in pathways associated with female reproduction and characteristics of oocytes maturation arrest and early embryo arrest by looking up MGI/PubMed database (New added Fig. 3b). However, in control group (unaffected siblings, public control group and ASD group), there were no significant enrichment in pathways associated with female reproduction. Notably, these pathways remained significantly enriched when we removed probands with recurrent miscarriage (Fig. 3; Additional file 1: Fig. S3). These results showed that specific enrichment of female reproductive pathways for DNM genes in our patient trios; (3) In addition to genes enriched in the female infertility pathway (listed in Fig. 3a), there are many other genes that may be associated with female infertility (new added Additional file 2: Table S5). It is hard to confirm the strong relationship between all de novo mutations and female infertility by using functional assay. So, we choose the top 1 gene TUBB4A as a representative to perform functional verification. Besides, we have added in discussion the probability of other candidates (UBQLN1, HTR2C and ZFPM2) with female infertility.

We have performed pathogenic prediction of DNMs in our previous manuscript, 53.19% of DNMs were predicted to be pathogenic (LoF + Damaging + Possible damaging) (Fig. 2b; Additional file 2: Table S2). Of note, all TUBA4A mutations as well as mutations in other three candidate genes with at least two DNMs are predicted to be pathogenic (Additional file 2: Table S2 and Table S7). Furthermore, some DNM genes were associated with female infertility by manually curating published literature and MGI datasets (Fig. 3b and Additional file 2: Table S4 and Table S5).

Based on these evidences, we believed that these DNM genes we identified play important roles in pathogenicity of female infertility.

Now the content has been revised as follows:

In the Results part (DNM genes are involved in pathways associated with female reproduction):

“To explore the functional role of DNMs, we performed GO analysis ($p < 0.01$) using stage-specific genes according to integrated expression clusters of different human early reproductive stages (Additional file 2: Table S3) and GO analysis is an established method in analyzing association relationship between DNMs and human diseases [10]”

“Selected Gene Ontology (GO) terms were integrated into 16 female reproductive stages (Fig. 3). We assigned GO terms to three main categories: meiosis, embryonic development and reproductive structure development. 11 DNM genes were enriched in pathways related to meiosis, such as meiotic cell cycle ($p = 1.85 \times 10^{-4}$), spindle organization ($p = 1.66 \times 10^{-3}$) and meiotic chromosome segregation ($p = 1.29 \times 10^{-4}$) (Additional file 1: Fig. S3). A total of 29 DNM genes were enriched in pathways related to embryonic development, such as in utero embryonic development ($p = 2.81 \times 10^{-4}$), regulation of translational initiation ($p = 3.81 \times 10^{-3}$) [23], regulation of histone modification ($p = 5.37 \times 10^{-4}$) [24-26] and positive regulation of Wnt signaling pathway ($p = 4.14 \times 10^{-6}$) [27-29] (Additional file 1: Fig. S3). 18 DNM genes were enriched in pathways related to reproductive structure development, such as urogenital system development ($p = 1.11 \times 10^{-3}$), genitalia development ($p = 1.50 \times 10^{-4}$), reproductive structure development ($p = 9.84 \times 10^{-3}$), female sex differentiation ($p = 3.24 \times 10^{-4}$) and development of primary female sexual characteristics ($p = 5.79 \times 10^{-3}$) (Additional file 1: Fig. S3). Overall, a group of DNM genes from our infertile parent-child trios was enriched in pathways of female reproductive processes (Additional file 2: Table S4) and these genes were closely related to female infertility with the characteristics of oocyte and early embryo defects by looking up MGI database and PubMed (Fig. 3b; Additional file 2: Table S5).”

Responses to Reviewer #2:

Reviewer #2: The paper is of broad interest to others in the field

Li et al present the first comprehensive analysis of rare de novo mutations (DNMs) detected by WES in infertile females and identified a set of 568 DNMs. The data suggested that the corresponding DNM genes were involved in the pathways associated with female reproductive processes. They also selected 9 genes for further evaluation, which included TUBA4A, UBQLN1, HTR2C, FRMD1, ZFPM2, ELFN2, UHRFBP1, ROBO2 and YLPM1. The data suggested that they were enriched in cohort with abnormalities in oocyte and early embryonic development compared to other published unrelated DNM datasets. The finding is novel and interesting. Further functional study illustrated that TUBA4A was highly expressed in human oocytes and weakly expressed in other somatic tissues. Mutant TUBA4A in mouse study demonstrated that TUBA4A may play an important role during human oocyte maturation and early embryo development. There are several issues need to be clarify:

Response: *We greatly appreciate this positive appraisal of our work and the encouraging comments.*

Major points:

1. DNMs are known to play a prominent role in human diseases, and the utilization of trio-based exome sequencing is a sound and reliable approach. What is the meaning of Parent-child trios here? I assume the Trios should be infertile females and their unaffected parents.

Response: *Parent-child trios means proband or sibling and her unaffected parents and was widely used in DNM studies^{25,35,41,45,46}. In this manuscript, “infertile females and their unaffected parents” were described as “infertile parent-child trios”. To make it clear, we have described this in more detail in the revised manuscript.*

2. Based on the method described, the criteria to filter the DNMs does it also include (MAF < 0.1%)? How many of the Candidate DNMs were verified by Sanger sequencing any possibility of low-level mosaicism among the parents?

Response: *Mutant allele frequency of all DNMs was lower than 0.01% in Asian populations and frequency was annotated by ANNOVAR software⁴⁷. Minor allele frequency of all DNMs were also pass filter by ANNOVAR software in Asian populations with threshold 0.1%. This threshold was widely used in DNMs analysis studies^{41,46}.*

In this study, 6 recurrent DNMs (13 trios: 3 TUBA4A, 2 UBQLN1, 2 HTR2C, 2 ZFPM2, 2 PDCH20 and 2 DGKZ) and 13 randomly selected DNMs (13 trios: 1 BUB1, 1 DAAM1, 1 MED19, 1 TPD52, 1 TTK, 1 MLH3, 1 LHX8, 1 CRELD1, 1 FRMD1, 1 C12orf29, 1 UHRF1BP1, 1 CCDC33 and 1 YLPM1) were verified by sanger sequencing (Additional file 1: Fig. S7). Totally, all 26 trios were verified by sanger sequencing and raw sanger sequencing data were upload into our GitHub code repository. For detection of low-level parental somatic mosaicism, we need collect more somatic tissues. In addition, DNA samples will be tested using PCR amplicon next-generation sequencing

(amplicon NGS) and droplet digital PCR (ddPCR). These processes make it more expensive and time-consuming. To date, only a few studies try to analyze low-level mosaicism in a large exome sequencing cohort^{48,49}. To make our results accurate, strict screen criteria were applied to identify DNMs in our manuscript: (1) allele frequency < 0.0001 in east Asian; (2) depth of proband/depth of parents > 0.3; (3) allele balance of proband > 0.2; (4) Genotyping quality score > 40; (5) all DNMs were verified using IGV software; (6) Randomly selected DNMs were verified by sanger sequencing. (7) Our analysis pipelines were widely used in other DNM studies^{35,40,41,46}. Based above reasons, we believed that our results are convincing.

Now the content has been revised as follows:

In the Methods part (Filtering and analysis of DNMs):

“Candidate DNMs were then verified by Sanger sequencing of PCR amplicons in participants who provided DNA and raw sanger sequencing data were upload into our GitHub code repository”

3. Table S1, I believe the Trio cohort is precious, for the recurrent miscarriage trios, what is the selection criteria? It seems that the DNMs are more towards oocyte maturation and early embryo development.

Response: *For sample collection, a complete medical evaluation was performed. Laboratory tests (e.g. probands’ semen examination, blood test), hormone tests and morphological examinations have become an integral part of clinical case selections for infertility and these procedures were performed by corresponding hospital and reproductive centers. Recurrent miscarriages have multiple etiologies, including chromosomal abnormalities, immune dysfunction, and various endocrine disorders. More than 90% were found to have at least one chromosomal abnormality in one or more cells in all blastomeres in preimplantation human embryos¹⁹. However, probands with recurrent miscarriage in our manuscript were diagnosed with primary infertility of unknown cause. So, the number of recurrent miscarriage trios was less than other two probands’ group. We thank reviewers for pointing out the oversight of our work. Although strict screening criteria for recurrent miscarriage trios are processed, it is difficult to ensure that the infertility in these probands is caused by female factors. And also based on the comment of the reviewer #1, we removed these probands and reanalyze the data in revised manuscript.*

4. Most of the known genetic findings towards TUBA4A is related to amyotrophic lateral sclerosis (ALS). It is very nice that the team also include the two reported mutations (p.R215C, p.R320H) associated with familial (ALS) to illustrate the differential function. Can they elaborate more regarding the possible mechanism of one gene contribute to different diseases in the discussion? Will this also explain why the rate of polar body extrusion and blastocyst formation are different between mutations?

Response: *We thank the reviewer for this positive assessment of our work and raising this very important question. Mutations in the same gene may result in different disease phenotypes due to the different genetic backgrounds of each individual^{50,51}. Firstly, Mutations in one gene may show incomplete penetrance or complete penetrance due to modifiers, epistatic interactions or suppressors present in the genome among different individuals; Secondly, this phenomenon may be*

caused by other factors, such as age, sex and environment⁵². For example, FLG mutations in ichthyosis vulgaris and atopic eczema show significant difference between European population and Asians⁵³; and patients with same mutations in NFI gene have huge heterogeneity in pathogenicity and clinical features⁵⁴. Additionally, background effects were verified in other disease caused by de novo mutations, such as congenital heart disease³⁶ and schizophrenia⁵⁵. In this study, all affected individuals with TUBA4A mutations only exhibited the phenotype of female infertility without any abnormalities in somatic tissues or organs (Additional file 2: Table S6). Firstly, TUBA4A mutations cause different phenotypes may due to individual's background effects. For example, TUBA4A show no significant enrichment in China ALS patients⁵⁶. Secondly, this might due to the different thresholds of TUBA4A amount required between oocytes and other tissues (Fig. 5c); Finally, different mutations in TUBA4A may cause different phenotype due to different pathogenicity of these mutations. In this study, mutations of TUBA4A cause different degrees of microtubule destabilization (Fig. 6c). Altogether, these TUBA4A mutation cause abnormality of oocyte maturation or embryo development.

Now the content has been revised as follows:

In the discussion part:

“Mutations in the same gene may result in different disease phenotypes due to the different genetic backgrounds of each individual.[52]. Firstly, Mutations in one gene may show incomplete penetrance or complete penetrance due to modifiers, epistatic interactions or suppressors present in the genome among different individuals; Secondly, this phenomenon may be caused by other factors, such as age, sex and environment [53-56]. Additionally, background effects were verified in other disease caused by de novo mutations, such as congenital heart disease [11] and schizophrenia [32]. One report showed that individuals with familial ALS had TUBA4A mutations based on exome sequencing data from large cohorts of European and American patients [36]. However, Li and colleagues found there were no TUBA4A causative mutations in 580 Chinese ALS patients, which suggested that TUBA4A may not be a significant genetic factor in Chinese ALS patients [57]. In this study, all affected individuals with TUBA4A mutations only exhibited the phenotype of female infertility without any abnormalities in somatic tissues or organs. Firstly, TUBA4A mutations cause different diseases may be due to individual's background effects. Secondly, this might due to the different thresholds of TUBA4A amount required between oocytes and other tissues (Fig. 5c); Finally, different mutations in TUBA4A may cause different phenotype due to different pathogenicity of these mutations. In this study, mutations of TUBA4A cause different degrees of microtubule destabilization (Fig. 6c). Altogether, our results showed that TUBA4A mutations cause abnormality of oocyte maturation or embryo development.”

Minor:

1. There are new single-cell profiling support TUBA4A transcriptomic changes during in vitro maturation of human oocytes <https://doi.org/10.1002/rmb2.12464>

Response: *We have added and cited this data in our manuscript. This data will make our work better (Fig. 3a; Additional file 1: Fig. S3).*

2. Within the DNMs how many are AD, AR or X-linked?

Response: *De novo mutations have extremely low mutation rates, 1.20×10^{-8} per nucleotide per generation²⁶. In this study, all DNMs were heterozygous. There were no autosomal recessive DNMs. Of these 481 DNMs, 16 DNMs were located on X chromosome (Additional file 2: Table S2).*

3. Ethics approval no ref. number was provided.

Response: *We have added Ethics approval ref. number in this manuscript [Ethics approval number: 162, 20161207, 2021-006 and JIAI E2021-33].*

References

- 1 Jiang, S. *et al.* The impact of blastomere loss on pregnancy and neonatal outcomes of vitrified-warmed Day3 embryos in single embryo transfer cycles. *Journal of Ovarian Research* **15**, 62, doi:10.1186/s13048-022-00997-z (2022).
- 2 Kuang, Y. *et al.* Medroxyprogesterone acetate is an effective oral alternative for preventing premature luteinizing hormone surges in women undergoing controlled ovarian hyperstimulation for in vitro fertilization. *Fertil Steril* **104**, 62-70.e63, doi:10.1016/j.fertnstert.2015.03.022 (2015).
- 3 Cummins, J. M. *et al.* A formula for scoring human embryo growth rates in in vitro fertilization: its value in predicting pregnancy and in comparison with visual estimates of embryo quality. *J In Vitro Fert Embryo Transf* **3**, 284-295, doi:10.1007/bf01133388 (1986).
- 4 Jiao, S. Y., Yang, Y. H. & Chen, S. R. Molecular genetics of infertility: loss-of-function mutations in humans and corresponding knockout/mutated mice. *Hum Reprod Update* **27**, 154-189, doi:10.1093/humupd/dmaa034 (2021).
- 5 Sha, Q.-Q. *et al.* Dynamics and clinical relevance of maternal mRNA clearance during the oocyte-to-embryo transition in humans. *Nature Communications* **11**, 4917, doi:10.1038/s41467-020-18680-6 (2020).
- 6 Chen, X. *et al.* Key role for CTCF in establishing chromatin structure in human embryos. *Nature* **576**, 306-310, doi:10.1038/s41586-019-1812-0 (2019).
- 7 Liu, H. B. *et al.* RNA-Binding Protein IGF2BP2/IMP2 is a Critical Maternal Activator in Early Zygotic Genome Activation. *Adv Sci (Weinh)* **6**, 1900295, doi:10.1002/advs.201900295 (2019).
- 8 Xu, Y. *et al.* Mutations in PADI6 Cause Female Infertility Characterized by Early Embryonic Arrest. *American journal of human genetics* **99**, 744-752, doi:10.1016/j.ajhg.2016.06.024 (2016).
- 9 Mu, J. *et al.* Mutations in NLRP2 and NLRP5 cause female infertility characterised by early embryonic arrest. *J Med Genet* **56**, 471-480, doi:10.1136/jmedgenet-2018-105936 (2019).
- 10 Zheng, W. *et al.* Homozygous Mutations in BTG4 Cause Zygotic Cleavage Failure and Female Infertility. *American journal of human genetics* **107**, 24-33, doi:10.1016/j.ajhg.2020.05.010 (2020).
- 11 Zhang, H. *et al.* Dominant mutations in CHK1 cause pronuclear fusion failure and zygote arrest that can be rescued by CHK1 inhibitor. *Cell Res*, doi:10.1038/s41422-021-00507-8 (2021).
- 12 Zhao, L. *et al.* Biallelic mutations in CDC20 cause female infertility characterized by abnormalities in oocyte maturation and early embryonic development. *Protein Cell* **11**, 921-927,

- doi:10.1007/s13238-020-00756-0 (2020).
- 13 Dong, J. *et al.* Novel biallelic mutations in MEI1: expanding the phenotypic spectrum to human embryonic arrest and recurrent implantation failure. *Hum Reprod* **36**, 2371-2381, doi:10.1093/humrep/deab118 (2021).
- 14 Wang, W. *et al.* FBXO43 variants in patients with female infertility characterized by early embryonic arrest. *Hum Reprod* **36**, 2392-2402, doi:10.1093/humrep/deab131 (2021).
- 15 Zeng, Y. *et al.* Bi-allelic mutations in MOS cause female infertility characterized by preimplantation embryonic arrest. *Hum Reprod* **37**, 612-620, doi:10.1093/humrep/deab281 (2022).
- 16 Chen, B. *et al.* Novel mutations and structural deletions in TUBB8: expanding mutational and phenotypic spectrum of patients with arrest in oocyte maturation, fertilization or early embryonic development. *Hum Reprod* **32**, 457-464, doi:10.1093/humrep/dew322 (2017).
- 17 Oud, M. S. *et al.* A de novo paradigm for male infertility. *Nature Communications* **13**, 154, doi:10.1038/s41467-021-27132-8 (2022).
- 18 Kaplanis, J. *et al.* Evidence for 28 genetic disorders discovered by combining healthcare and research data. *Nature*, doi:10.1038/s41586-020-2832-5 (2020).
- 19 Larsen, E. C., Christiansen, O. B., Kolte, A. M. & Macklon, N. New insights into mechanisms behind miscarriage. *BMC Med* **11**, 154, doi:10.1186/1741-7015-11-154 (2013).
- 20 Shen, L. *et al.* Effects of early age at natural menopause on coronary heart disease and stroke in Chinese women. *Int J Cardiol* **241**, 6-11, doi:10.1016/j.ijcard.2017.03.127 (2017).
- 21 Muka, T. *et al.* Association of Age at Onset of Menopause and Time Since Onset of Menopause With Cardiovascular Outcomes, Intermediate Vascular Traits, and All-Cause Mortality: A Systematic Review and Meta-analysis. *JAMA Cardiol* **1**, 767-776, doi:10.1001/jamacardio.2016.2415 (2016).
- 22 Shen, L. *et al.* Association between earlier age at natural menopause and risk of diabetes in middle-aged and older Chinese women: The Dongfeng-Tongji cohort study. *Diabetes Metab* **43**, 345-350, doi:10.1016/j.diabet.2016.12.011 (2017).
- 23 Carson, S. A. & Kallen, A. N. Diagnosis and Management of Infertility: A Review. *Jama* **326**, 65-76, doi:10.1001/jama.2021.4788 (2021).
- 24 Tuddenham, S., Hamill, M. M. & Ghanem, K. G. Diagnosis and Treatment of Sexually Transmitted Infections: A Review. *Jama* **327**, 161-172, doi:10.1001/jama.2021.23487 (2022).
- 25 Willsey, A. J. *et al.* De Novo Coding Variants Are Strongly Associated with Tourette Disorder. *Neuron* **94**, 486-499 e489, doi:10.1016/j.neuron.2017.04.024 (2017).
- 26 Kong, A. *et al.* Rate of de novo mutations and the importance of father's age to disease risk. *Nature* **488**, 471-475, doi:10.1038/nature11396 (2012).
- 27 Acuna-Hidalgo, R., Veltman, J. A. & Hoischen, A. New insights into the generation and role of de novo mutations in health and disease. *Genome Biol* **17**, 241, doi:10.1186/s13059-016-1110-1 (2016).
- 28 Lynch, M. Rate, molecular spectrum, and consequences of human mutation. *Proc Natl Acad Sci U S A* **107**, 961-968, doi:10.1073/pnas.0912629107 (2010).
- 29 Lynch, M. Evolution of the mutation rate. *Trends Genet* **26**, 345-352, doi:10.1016/j.tig.2010.05.003 (2010).
- 30 Rivière, J. B. *et al.* De novo germline and postzygotic mutations in AKT3, PIK3R2 and PIK3CA cause a spectrum of related megalencephaly syndromes. *Nat Genet* **44**, 934-940,

- doi:10.1038/ng.2331 (2012).
- 31 Kordus, R. J. & LaVoie, H. A. Granulosa cell biomarkers to predict pregnancy in ART: pieces
to solve the puzzle. *Reproduction* **153**, R69-r83, doi:10.1530/rep-16-0500 (2017).
- 32 Parfitt, D.-E. *et al.* Defining a clinical validity framework for pharmacogenomic biomarkers of
IVF treatment response and outcomes. *Fertil. Steril.* **112**, e260,
doi:https://doi.org/10.1016/j.fertnstert.2019.07.784 (2019).
- 33 Kyrönlahti, A. *et al.* GATA4 deficiency impairs ovarian function in adult mice. *Biol Reprod* **84**,
1033-1044, doi:10.1095/biolreprod.110.086850 (2011).
- 34 van den Bergen, J. A. *et al.* Analysis of variants in GATA4 and FOG2/ZFPM2 demonstrates
benign contribution to 46,XY disorders of sex development. *Mol Genet Genomic Med* **8**, e1095,
doi:10.1002/mgg3.1095 (2020).
- 35 Satterstrom, F. K. *et al.* Large-Scale Exome Sequencing Study Implicates Both Developmental
and Functional Changes in the Neurobiology of Autism. *Cell* **180**, 568-584 e523,
doi:10.1016/j.cell.2019.12.036 (2020).
- 36 Jin, S. C. *et al.* Contribution of rare inherited and de novo variants in 2,871 congenital heart
disease probands. *Nat Genet* **49**, 1593-1601, doi:10.1038/ng.3970 (2017).
- 37 Chong, J. X. *et al.* The Genetic Basis of Mendelian Phenotypes: Discoveries, Challenges, and
Opportunities. *American journal of human genetics* **97**, 199-215,
doi:10.1016/j.ajhg.2015.06.009 (2015).
- 38 Nagata, Y. *et al.* Germline loss-of-function SAMD9 and SAMD9L alterations in adult
myelodysplastic syndromes. *Blood* **132**, 2309-2313, doi:10.1182/blood-2017-05-787390 (2018).
- 39 Hengel, H. *et al.* Bi-allelic loss-of-function variants in BCAS3 cause a syndromic
neurodevelopmental disorder. *American journal of human genetics* **108**, 1069-1082,
doi:10.1016/j.ajhg.2021.04.024 (2021).
- 40 Sadler, B. *et al.* Rare and de novo coding variants in chromodomain genes in Chiari I
malformation. *The American Journal of Human Genetics*, doi:10.1016/j.ajhg.2020.12.001
(2020).
- 41 Bishop, M. R. *et al.* Genome-wide Enrichment of De Novo Coding Mutations in Orofacial Cleft
Trios. *American journal of human genetics* **107**, 124-136, doi:10.1016/j.ajhg.2020.05.018
(2020).
- 42 Zhao, L. *et al.* Heterozygous loss-of-function variants in LHX8 cause female infertility
characterized by oocyte maturation arrest. *Genet Med* **24**, 2274-2284,
doi:10.1016/j.gim.2022.07.027 (2022).
- 43 Kataoka, M. *et al.* Exome sequencing for bipolar disorder points to roles of de novo loss-of-
function and protein-altering mutations. *Mol Psychiatry* **21**, 885-893, doi:10.1038/mp.2016.69
(2016).
- 44 Hamanaka, K. *et al.* Large-scale discovery of novel neurodevelopmental disorder-related genes
through a unified analysis of single-nucleotide and copy number variants. *Genome Med* **14**, 40,
doi:10.1186/s13073-022-01042-w (2022).
- 45 Sanna-Cherchi, S. *et al.* Exome-wide Association Study Identifies GREB1L Mutations in
Congenital Kidney Malformations. *American journal of human genetics* **101**, 789-802,
doi:10.1016/j.ajhg.2017.09.018 (2017).
- 46 Qiao, L. *et al.* Rare and de novo variants in 827 congenital diaphragmatic hernia probands
implicate LONP1 as candidate risk gene. *American journal of human genetics* **108**, 1964-1980,

- doi:10.1016/j.ajhg.2021.08.011 (2021).
- 47 Wang, K., Li, M. & Hakonarson, H. ANNOVAR: functional annotation of genetic variants from
high-throughput sequencing data. *Nucleic Acids Res* **38**, e164, doi:10.1093/nar/gkq603 (2010).
- 48 Domogala, D. D. *et al.* Detection of low-level parental somatic mosaicism for clinically relevant
SNVs and indels identified in a large exome sequencing dataset. *Human Genomics* **15**, 72,
doi:10.1186/s40246-021-00369-6 (2021).
- 49 Hu, P. *et al.* Low-level parental mosaicism affects the recurrence risk of holoprosencephaly.
Genetics in Medicine **21**, 1015-1020, doi:10.1038/s41436-018-0261-8 (2019).
- 50 Chow, C. Y. Bringing genetic background into focus. *Nature Reviews Genetics* **17**, 63-64,
doi:10.1038/nrg.2015.9 (2016).
- 51 Sackton, T. B. & Hartl, D. L. Genotypic Context and Epistasis in Individuals and Populations.
Cell **166**, 279-287, doi:10.1016/j.cell.2016.06.047 (2016).
- 52 Fournier, T. & Schacherer, J. Genetic backgrounds and hidden trait complexity in natural
populations. *Curr Opin Genet Dev* **47**, 48-53, doi:10.1016/j.gde.2017.08.009 (2017).
- 53 Akiyama, M. FLG mutations in ichthyosis vulgaris and atopic eczema: spectrum of mutations
and population genetics. *Br J Dermatol* **162**, 472-477, doi:10.1111/j.1365-2133.2009.09582.x
(2010).
- 54 Pasmant, E., Vidaud, M., Vidaud, D. & Wolkenstein, P. Neurofibromatosis type 1: from
genotype to phenotype. *J Med Genet* **49**, 483-489, doi:10.1136/jmedgenet-2012-100978 (2012).
- 55 Fromer, M. *et al.* De novo mutations in schizophrenia implicate synaptic networks. *Nature* **506**,
179-184, doi:10.1038/nature12929 (2014).
- 56 Li, J. *et al.* TUBA4A may not be a significant genetic factor in Chinese ALS patients. *Amyotroph
Lateral Scler Frontotemporal Degener* **17**, 148-150, doi:10.3109/21678421.2015.1074705
(2015).

Second round of review

Reviewer 1

The authors have extensively revised the original manuscript based on my comments and I am very happy with the improvements. I have no further comments.

Reviewer 2

All issues have been well addressed.